# Chromatin dynamics and the role of G9a in gene regulation and enhancer silencing during early mouse development

Jan J Zylicz[1,2,3], Sabine Dietmann[3], Ufuk Günesdogan[1,2], Jamie A Hackett[1,2], Delphine Cougot[1,2], Caroline Lee[1,2], M Azim Surani[1,2]*

[1]Wellcome Trust/Cancer Research United Kingdom Gurdon Institute, University of Cambridge, Cambridge, United Kingdom; [2]Department of Physiology, Development and Neuroscience, University of Cambridge, Cambridge, United Kingdom; [3]Wellcome Trust/Medical Research Council Stem Cell Institute, University of Cambridge, Cambridge, United Kingdom

**Abstract** Early mouse development is accompanied by dynamic changes in chromatin modifications, including G9a-mediated histone H3 lysine 9 dimethylation (H3K9me2), which is essential for embryonic development. Here we show that genome-wide accumulation of H3K9me2 is crucial for postimplantation development, and coincides with redistribution of enhancer of zeste homolog 2 (EZH2)-dependent histone H3 lysine 27 trimethylation (H3K27me3). Loss of G9a or EZH2 results in upregulation of distinct gene sets involved in cell cycle regulation, germline development and embryogenesis. Notably, the H3K9me2 modification extends to active enhancer elements where it promotes developmentally-linked gene silencing and directly marks promoters and gene bodies. This epigenetic mechanism is important for priming gene regulatory networks for critical cell fate decisions in rapidly proliferating postimplantation epiblast cells.

*For correspondence: a.surani@gurdon.cam.ac.uk

**Competing interests:** The authors declare that no competing interests exist.

## Introduction

Early mammalian development progresses through a series of landmark events that are regulated by transcriptional and epigenetic mechanisms. Accordingly, establishment of the pluripotent inner cell mass (ICM) in E3.5 blastocysts is linked with global DNA hypomethylation (*Figure 1A*) (*Hackett et al., 2013a*; *Smith et al., 2012*; *Wang et al., 2014*). Subsequent development of the ICM leads to the formation of primed postimplantation epiblast cells by E5.5–6.25, which are poised to initiate lineage-specification. This developmental transition is accompanied by epigenetic programming, including genome-wide de novo DNA methylation, and potentially accumulation of histone H3 lysine 9 dimethylation (H3K9me2) and redistribution of histone H3 lysine 27 trimethylation (H3K27me3) (*Figure 1A*) (*Borgel et al., 2010*; *Leitch et al., 2013*; *Marks et al., 2012*). Methyltransferase enzymes, G9a (encoded by *Ehmt2*) and enhancer of zeste homolog 2 (EZH2), respectively, are responsible for the establishment of these chromatin modifications (*O'Carroll et al., 2001*; *Tachibana et al., 2002*; *2005*). However, the precise contribution of epigenetic programming for setting and regulating the transcriptional circuitry in early development remains to be fully elucidated (*Li et al., 1992*; *O'Carroll et al., 2001*; *Okano et al., 1999*; *Tachibana et al., 2002*).

The involvement of specific enhancer elements in these events is of particular interest as they undergo rapid epigenetic setting, for example, by becoming activated or poised with histone H3 lysine 27 acetylation (H3K27ac) and H3K27me3 modifications, respectively (*Creyghton et al., 2010*; *Heintzman et al., 2009*; *Nord et al., 2013*; *Rada-Iglesias et al., 2011*; *Zentner et al., 2011*). Similarly, the transition from naïve embryonic stem cells (ESCs) to epiblast-like cells (EpiLCs) and epiblast

**eLife digest** The genome contains full instructions for the development of the whole organism. The genes within the genome encode for all the proteins, but specific genes are selected to be active at the appropriate time. For this reason, there are mechanisms that can turn the genes on and off as and when required. One such mechanism is called methylation, in which a chemical group called a methyl tag is added to either the DNA or to histone proteins. DNA wraps around histone proteins to form a structure called chromatin. When histones are tagged with a methyl group, they become closely packed, and the resulting compaction of the chromatin around a gene inactivates that gene. For gene activation, the methyl tag is replaced by another tag called acetyl, which allows the chromatin to de-compact and become more accessible.

Soon after fertilisation, most of the methyl tags an embryo inherits from its parents are removed and a group of stem cells is established. An opposite process takes place once the embryo "nests" in the uterus. Two enzymes called G9a and EZH2 add methyl tags to specific residues on specific histones to regulate the expression of genes at the time when cells begin to decide what tissue to become. Zylicz et al. have now investigated how these enzymes contribute to the changes in chromatin packing that occur during early mouse development that is essential for the progression of normal development.

In mouse embryos that lacked the G9a enzyme, Zylicz et al. found that specific genes were inappropriately and prematurely activated early in development, including those generally involved in regulating 'reproduction' and 'cell death'. Similarly, embryos that lacked the EZH2 enzyme experienced the early and inappropriate activation of a different set of developmentally important genes. In addition, switches called enhancers control gene expression, and Zylicz et al. found that these regulators are also turned off by histone modifications made by the G9a enzyme. Thus, these findings provide important insights on how genes are regulated during a critical period of mouse development, when cells prepare to acquire their specific identities. This should lead to further work on the role of G9a in the hours after fertilisation; that is before the 'programming' events described here.

stem cells (EpiSCs) is accompanied by rapid changes in enhancer usage (*Buecker et al., 2014*; *Factor et al., 2014*). These in vitro states are equivalent to the ICM, E6.25 epiblast and the primitive streak, respectively (*Figure 1A*) (*Boroviak et al., 2014*; *Hayashi et al., 2011*; *Kojima et al., 2014*). Similar events likely occur in postimplantation epiblast in vivo when cell cycle shortening is linked with extensive transcriptional and epigenetic alterations, in preparation for key cell fate decisions, including specification of somatic and germline fates (*Borgel et al., 2010*; *Snow and Bennett, 1978*). The modifications of enhancers in this context may be of crucial importance for ensuring appropriate response to the ongoing developmental cues.

In this study, we have focused on the contribution of G9a-mediated H3K9me2 and EZH2-dependent H3K27me3 to early mouse development. We found that during the formation of postimplantation epiblast, there is a dramatic increase in H3K9me2 levels and a concomitant H3K27me3 redistribution. These events are necessary for repression of a distinct set of genes, including regulators of the germline, cell cycle, apoptosis, and development. The rapid acquisition of H3K9me2 extends to key enhancer elements, thereby reinforcing their repression. We propose that such epigenetic programming of epiblast primes a specific gene regulatory network, which is a necessary prerequisite for embryogenesis.

## Results

### Epigenetic programming regulates growth and development of the embryo

First, we investigated the dynamics of epigenetic programming of repressive H3K9me2 during early mouse development. Immunofluorescence (IF) analysis of E3.5 and E5.5 embryos revealed significant enrichment of H3K9me2 in the epiblast of postimplantation embryos (*Figure 1B,E*). Accumulation of

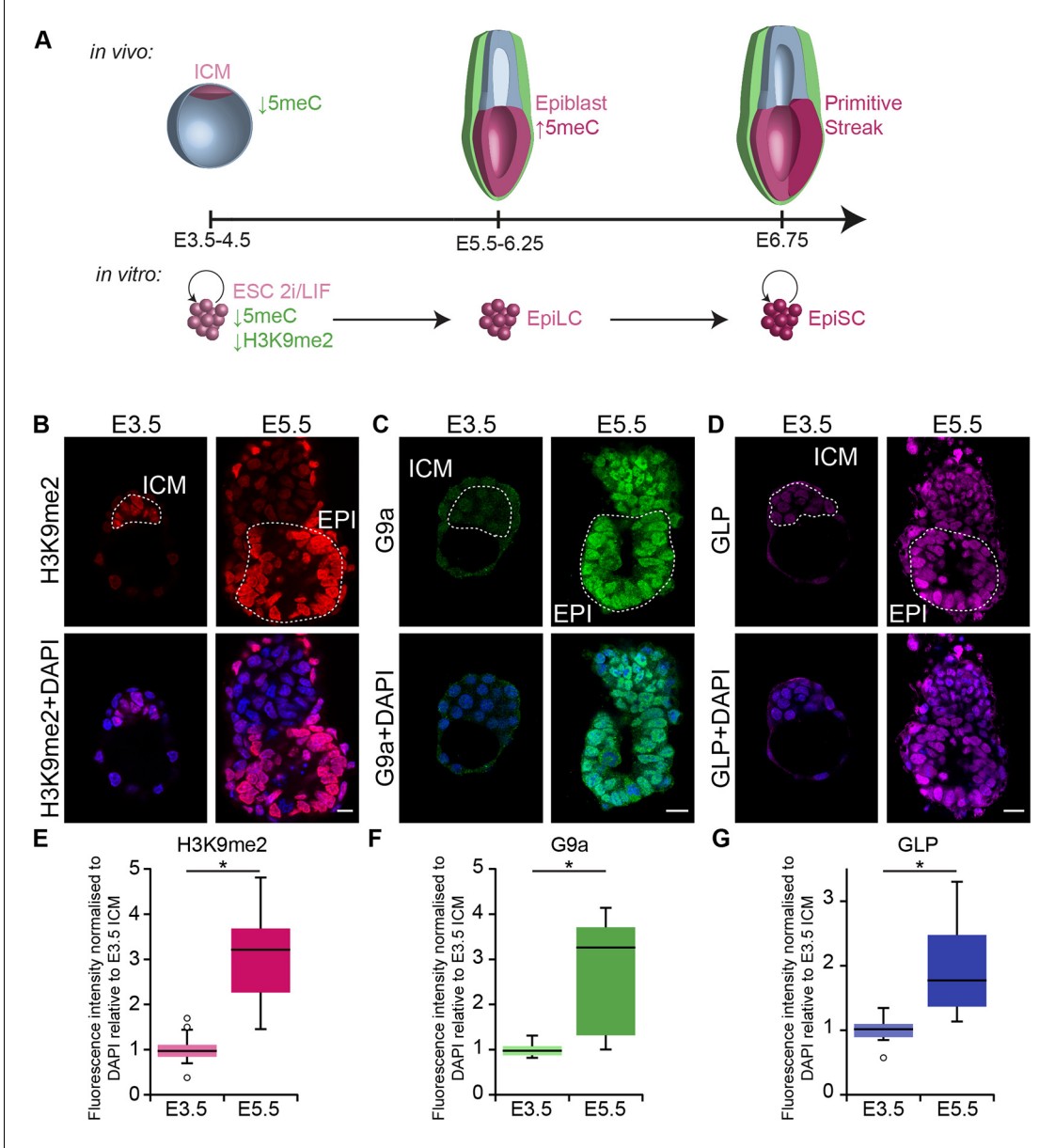

**Figure 1.** G9a-dependent programming occurs at implantation. (**A**) Schematic of early mouse development and their in vitro equivalents. Genome-wide DNA demethylation after fertilisation leads to an epigenetic basal state with low 5meC in ICM of blastocysts. Shortly after implantation, the epiblast cells undergo epigenetic programming, which includes de novo DNA methylation. By E6.25, the epiblast is primed for somatic development, while being competent for germline specification. Gastrulation follows at E6.75. Naïve ESCs, primed EpiLCs and EpiSCs represent different stages of in vivo development. ESCs grown in 2i/LIF medium resemble the ICM, while EpiLCs, induced from ESCs after 48 h in response to FGF2 and Activin A, are equivalent to epiblast. Their prolonged culture results in EpiSCs, which are reminiscent of the anterior primitive streak (*Kojima et al., 2014*). (**B–D**) Whole-mount IF staining for H3K9me2 (**B**), G9a (**C**) and GLP (**D**) in E3.5 and E5.5 embryos. Dotted line shows the ICM, and the EPI. It is noteworthy that a single confocal plane is shown to maintain original IF intensity. For anti-G9a staining of E5.5 embryo, visceral endoderm was removed to reduce the background signal (scale bar = 20 μm). (**E–G**) Box plots showing IF signal quantification for H3K9me2 (**E**), G9a (**F**) and GLP (**G**). Data shows IF intensity normaliseormalized to DAPI for individual ICM or epiblast cells. At least 3 embryos and 20 cells were quantified for each time point. (*p<0.05 in Wilcoxon rank sum test). DAPI: 4′,6-diamidino-2-phenylindole; EPI: epiblast; EpiLCs: epiblast-like cells; ESCs: embryonic stem cells; FGF2: fibroblast growth factor 2; GLP: G9a-like protein; ICM: inner cell mass; IF: immunofluorescence; 2i/LIF: two-inhibitor/leukemia inhibitory factor.

this modification, which is dependent on G9a and its binding partner G9a-like protein (GLP), coincides with increased levels of the enzymes (*Figure 1C,D,F,G*). To address the function of H3K9me2, we examined the consequences of *Ehmt2* deletion (*Ehmt2*$^{-/-}$). Loss of G9a by E6.5 resulted in

reduced levels of H3K9me2 modification, and an increase in apoptotic and non-proliferative cells as judged by IF staining for cleaved Caspase 3 and Ki67, respectively (*Figure 2—figure supplement 1A,C,D*). These changes led to a developmental delay of mutant embryos by E7.5 (*Figure 2A,B*), consistent with previous reports (*Tachibana et al., 2002*; *Yamamizu et al., 2012*).

To gain further insight into the underlying causes of the phenotype, we performed RNA sequencing (RNA-seq) on individual *Ehmt2*$^{-/-}$ epiblasts at E6.25. This revealed misregulation of 180 genes, of which 147 (~82%) are upregulated (*Figure 2C*, *Figure 2—source data 1*) (Log2[fold change[FC]] >1.4, p-value<0.05). Among the upregulated genes, ~27% of them were located on the X chromosome (p-value=1.6 $\times$ 10$^{-15}$), notably in clusters of *Xlr*, *Rhox* and *Mage-a* genes. Consistent with the known functions of these clusters, we found significant enrichment of gene ontology (GO) terms linked to hematopoiesis, sexual reproduction, and regulation cell proliferation (*Figure 2D*, *Figure 2— figure supplement 1B*, *Figure 2—source data 2*). To validate these findings, we analysed individual E6.25 epiblast cells by single cell real-time quantitative polymerase chain reaction (RT-qPCR). A significant proportion of *Ehmt2*$^{-/-}$ cells showed upregulation of the cyclin-dependent kinase inhibitor *Cdkn1a*, and late germline markers *Asz1* and *Rhox5* but not *Pou5f1* (coding for OCT4) or *Nanog* (*Figure 2E,F*, *Figure 2—figure supplement 2A,B*). This indicates that, contrary to a previous report (*Yamamizu et al., 2012*), the phenotypic effects cannot be attributed to a delayed exit from naïve pluripotency. Furthermore, loss of G9a did not abrogate the establishment of a population of primordial germ cells (PGCs), as judged by the expression of AP2γ and OCT4, key germline regulators (*Figure 2—figure supplement 2C,D*). These observations show that G9a promotes growth of the embryo by repressing apoptotic and late germline genes, but it does not affect the exit from naïve pluripotency and establishment of the PGC lineage.

Next, we examined the consequences of loss of *Ezh2* and thus of the H3K27me3 modification, which likely undergoes significant redistribution during epiblast development (*Marks et al., 2012*). For this reason, we performed RNA-seq on individual E6.25 epiblasts lacking EZH2 (*Ezh2*$^{-/-}$). We found upregulation of 165, and downregulation of 24 transcripts (Log2(FC)>1.4, p-value<0.05) (*Figure 2—figure supplement 3A,B*, *Figure 2—source data 3*), among which were homeobox and gastrulation-related genes, including *Hoxd13* and *Lefty2*, but pluripotency regulators such as *Nanog* and *Pou5f1* were not affected (*Figure 2—figure supplement 3B,C*, *Figure 2—source data 4*). Importantly, we only found five significantly upregulated genes that were shared between *Ehmt2*$^{-/-}$ and *Ezh2*$^{-/-}$ embryos. Thus, G9a and EZH2 appear to stabilise silencing of distinct sets of germline, proliferation and developmental regulators, but neither of them has an effect on the pluripotency transcription programme in postimplantation embryos.

## H3K9me2 and H3K27me3 are associated with distinct repressive chromatin states in vivo

To understand the roles of H3K9me2 and H3K27me3 modifications during the transition from naïve pluripotency in the ICM of blastocysts to a primed pluripotent state in postimplantation embryos, we investigated the genome-wide distribution of these modifications. For this purpose, we optimised a low cell number chromatin immunoprecipitation with sequencing (lcChIP-seq) protocol to analyse ~25,000 pregastrulation E6.25 epiblast cells in two biological replicates (*Figure 3—figure supplement 1A–C*)(*Ng, et al., 2013*). We intersected this information with our RNA-seq data and with the published whole genome bisulfite sequencing (WGBSeq) datasets (*Seisenberger et al., 2012*). This enabled us to generate a comprehensive overview of the epigenetic and transcriptional state of primed pluripotent epiblast cells in vivo.

The enrichment of H3K9me2 and H3K27me3 modifications in E6.25 epiblast is associated with low and high CpG content, respectively (*Figure 3—figure supplement 2A*). This is also the case in ESCs cultured in conventional media with serum (sESC) (*Lienert et al., 2011*; *Wen et al., 2009*). By contrast, naïve ESCs grown in 2i/LIF (2i/LIF ESCs) show spreading of H3K27me3 outside the CpG dense loci (*Marks et al., 2012*). Thus, there is redistribution of H3K27me3 in E6.25 epiblast, relative to both naïve ESCs and possibly ICM in vivo. The association of H3K9me2 and H3K27me3 modifications on promoters is mutually exclusive, since only 0.3% of them are enriched for both marks (*Figure 3A*, anticorrelation with Chi$^2$ p-value=0.0024). These differences are in line with H3K9me2 and H3K27me3 being linked to high and low 5-methylcytosine (5meC) levels, respectively (*Figure 3— figure supplement 2B*). Nonetheless, despite marking distinct chromatin regions, both H3K9me2 and H3K27me3 are linked to transcriptional repression (*Figure 3B*). Notably, this gene repression is

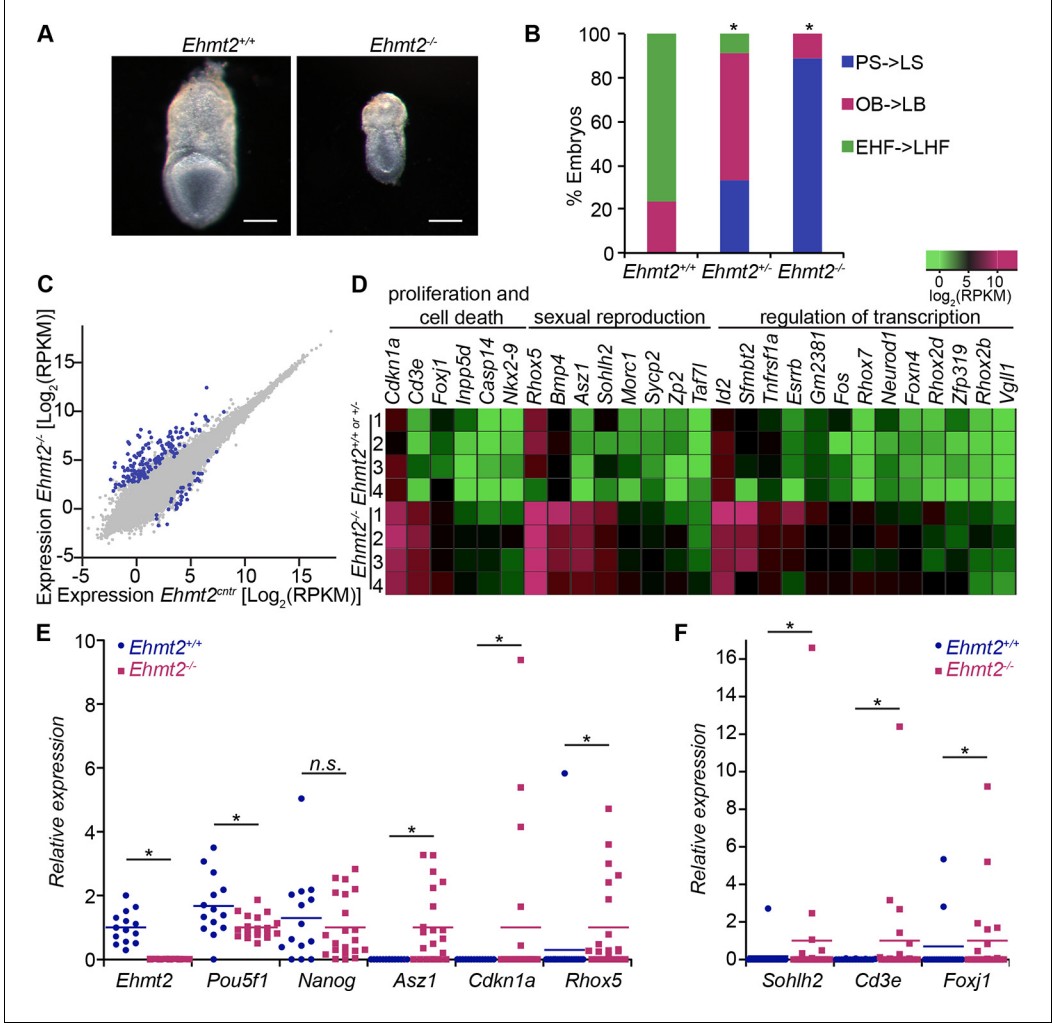

**Figure 2.** G9a represses germline and proliferation-related genes in the postimplantation epiblast. (A,B) bright-field images of *Ehmt2$^{+/+}$* and *Ehmt2$^{-/-}$* embryos at E7.5 (A) (scale bar = 0.1 mm). At least nine embryos of each type were staged (B) (*Chi$^2$ test p-value= <0.05). (C) Scatter plot showing transcript expression levels in *Ehmt2$^{+/+or +/-}$* and *Ehmt2$^{-/-}$* E6.25 epiblast. Blue points are differentially expressed genes (Log2RPKM>1, p-value<0.05, Log2(FC)>1.4). Shown is the geometric average from four biological replicates. (D) Heatmap showing expression of selected genes from enriched GO categories. (E,F) Single-cell RT-qPCR validation of RNA-seq performed on individual epiblast cells isolated from E6.25 *Ehmt2$^{+/+}$* or *Ehmt2$^{-/-}$* embryos (minimum 2 embryos and 14 cells). Dot plots show levels of *Ehmt2*, pluripotency (*Nanog, Pou5f1*), germline (*Asz1, Rhox5, Sohlh2*) and proliferation regulators (*Cdkn1a, Cd3e, Foxj1*). Expression is normalised to *Arbp* and relative to average in *Ehmt2$^{-/-}$* and for *Ehmt2* relative to *Ehmt2$^{+/+}$*. Statistical significance was calculated using Wilcoxon rank sum test for *Pou5f1* and *Nanog*, where majority of WT and KO cells show detectable expression. For remaining genes a Chi$^2$ test was used. (*p-value<0.05). Also see *Figure 2—source data 1–4* and *Figure 2—figure supplement 1–3*. LHF: late head fold; EHF: early head fold; LB: late allantoic bud; OB: no allantoic bud; LS: late streak; PS: pre-streak. RT-qPCR: real-time quantitative polymerase chain reaction; RNA-seq: RNA sequencing; WT: wild-type: KO: knockout; GO: gene ontology; FC: fold change.

The following source data and figure supplements are available for figure 2:

**Source data 1.** List of differentially expressed genes in E6.25 *Ehmt2$^{-/-}$* epiblast from RNA-seq analysis data is based on four individual *Ehmt2$^{-/-}$* and control (*Ehmt2$^{+/+}$* or *Ehmt2$^{+/-}$*) epiblasts.

**Source data 2.** List of enriched GO terms in genes upregulated in E6.25 *Ehmt2$^{-/-}$* epiblast GO term enrichment for biological processes was calculated using DAVID software with minimum five genes in a category and EASE p-value<0.05.

**Source data 3.** List of differentially expressed genes in E6.25 *Ezh2$^{-/-}$* epiblast from RNA-seq analysis data is based on 3 individual *Ezh2$^{-/-}$* epiblasts and 3 individual control (*Ezh2$^{+/+}$* or *Ezh2$^{+/-}$*) epiblasts.

*Figure 2 continued on next page*

*Figure 2 continued*

**Source data 4.** List of enriched GO terms in genes upregulated in E6.25 *Ezh2*$^{-/-}$ epiblast GO term enrichment for biological processes was calculated using DAVID software with minimum five genes in a category and EASE p-value<0.05.
**Figure supplement 1.** Epigenetic programming by G9a in postimplantation epiblast.
**Figure supplement 2.** Loss of G9a does not affect the exit from pluripotency and germline specification.
**Figure supplement 3.** EZH2 represses multiple developmental regulators in vivo.

correlated with histone modification enrichment at promoters as well as in gene bodies. The H3K9me2 modification in gene bodies could impede transcriptional elongation, splicing, or activity of regulatory elements (*Allo et al., 2009*). Our evidence suggests that H3K9me2 and H3K27me3 modifications in vivo are linked to distinct repressive chromatin states. We confirmed this by means of self-organizing maps, which cluster promoters and gene bodies based on similarity of their cumulative epigenetic signature with respect to transcriptional activity (*Figure 3C*)(*Wehrens and Buydens, 2007*).

To gain insight into the epigenetic regulation of developmental progression from naïve to primed pluripotent cells in vivo, we integrated our dataset from E6.25 epiblasts with RNA-seq of E3.5 ICM (ERP005749) (*Boroviak et al., 2014*). First, we identified genes that become robustly activated or repressed in E6.25 epiblasts relative to ICM (*Figure 3D*) (Log2(RPKM)<4, p-value<0.05, Log2(FC) >1). These genes generally corresponded to the expected developmental progression. For example, the transcripts that become silenced by the postimplantation stage (E6.25) are enriched for GO terms such as 'blastocyst formation' and 'STAT (Signal Transducer and Activator of Transcription) signalling regulation' (*Figure 3F*). These repressed genes are generally enriched for H3K9me2 or H3K27me3, especially when they have high or intermediate CpG density, respectively (*Figure 3E*, *Figure 3—figure supplement 3A*). Moreover, consistent with global DNA hypomethylation of the ICM, there is transient expression of 5meC-sensitive germline genes at E3.5 that are subsequently repressed by E6.25. On the other hand, genes activated in E6.25 epiblast include 'neural tube' and 'polarized epithelium' genes (*Figure 3F*). Finally, both activated and repressed genes are enriched for regulators of transcription, embryonic development and metabolic processes (*Figure 3F*). This analysis reveals that the transcriptional changes between the naïve state associated with the ICM, and the primed state of postimplantation epiblast in vivo reflect the dramatic alterations in signalling, morphology, and metabolism occurring during development.

Next, we focused on genes that are repressed upon implantation and accumulate H3K9me2 or H3K27me3. GO term analysis revealed preferential acquisition of H3K27me3 on genes associated with transcriptional regulation, embryonic organ development, blastocyst formation, and metabolism (*Figure 3G*). On the other hand, meiotic genes and those involved in immune responses were more likely targeted by H3K9me2. This analysis reveals that different functional pathways are inactivated via establishment of distinct epigenetic states.

## EpiSCs show aberrant epigenetic state of germline genes

Despite being derived from postimplantation embryos, EpiSCs have reduced competence for the germline fate (*Hayashi and Surani, 2009*). To investigate this, we directly compared the epigenetic states of E6.25 epiblast and EpiSCs. Global and metagene analysis revealed little differences in the distribution of H3K9me2 and H3K27me3 between E6.25 epiblast and EpiSCs (*Figure 4A,D*, *Figure 4—figure supplement 1A*). Similarly, over 97% of genes marked by H3K9me2 showed enrichment in both cell types (*Figure 4B,C*). On the other hand, H3K27me3 underwent a significant rearrangement in EpiSCs when compared with in vivo epiblast cells, with genes regulating meiosis and bone development preferentially losing H3K27me3 at promoters (*Figure 4E,F,H*). We reasoned that this might be associated with DNA hypermethylation of such elements since H3K27me3 mark is anti-correlated with 5meC (*Figure 3—figure supplement 2B*). Indeed, our whole genome bisulfite sequencing (WGBSeq) of EpiSCs revealed a ~10% increase in global DNA methylation levels (*Figure 4G*). More specifically, promoters that had lost H3K27me3 acquired significantly more 5meC

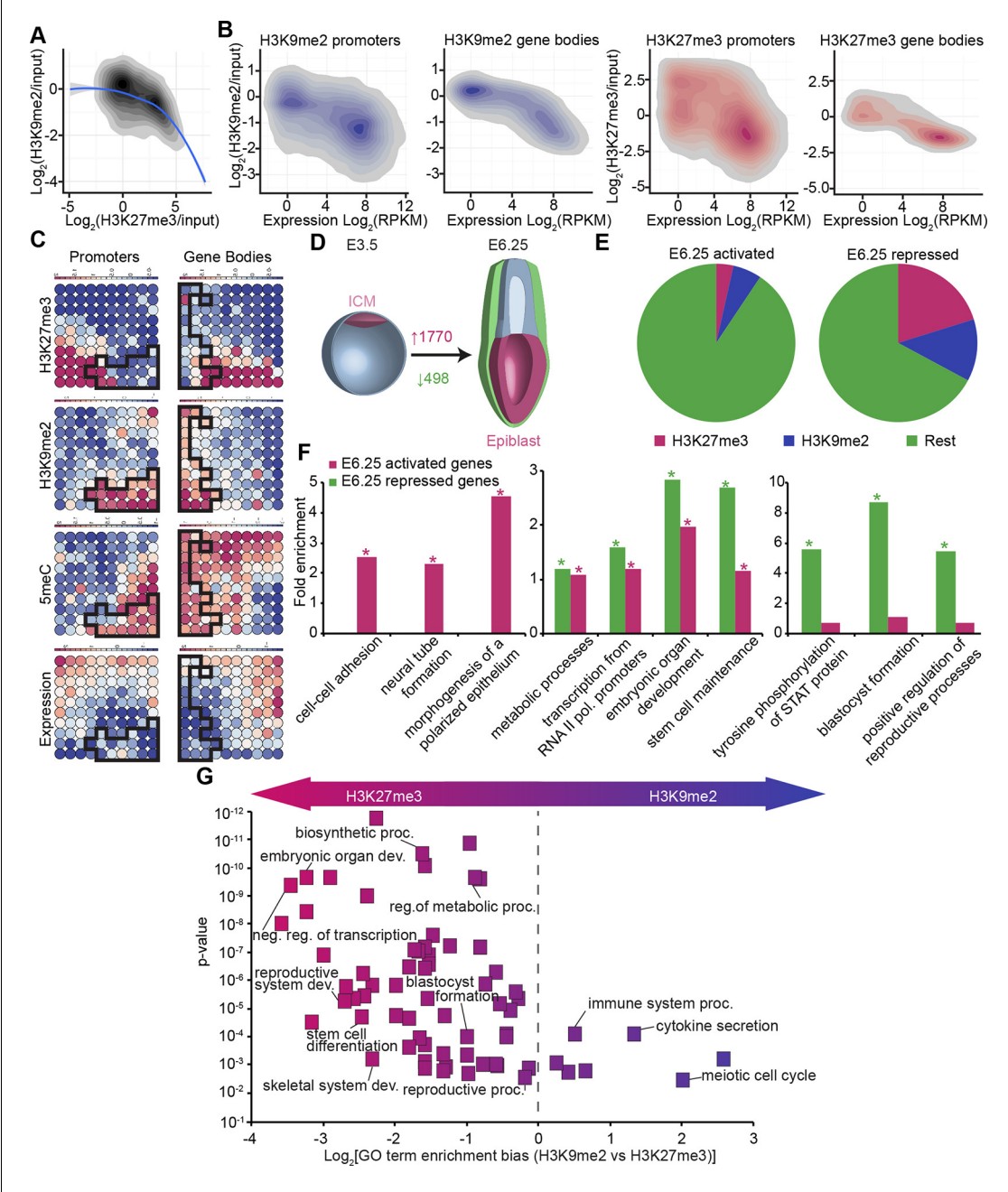

**Figure 3.** In vivo lcChip-seq from E6.25 epiblast reveals distinct epigenetic state of primed pluripotent cells. (**A**) Density contour plot showing the relationship between H3K9me2 and H3K27me3 enrichment. Shown are all promoters associated with genes repressed in the epiblast. (**B**) Density contour plots showing correlation between H3K9me2 (blue) and H3K27me3 (red) at promoters (left panels) and gene bodies (right panels), with transcriptional activity in epiblast. (**C**) Unbiased clustering of promoters (left panels) and gene bodies (right panels) based on their cumulative epigenetic and transcriptional signature in epiblast. Analysis was performed using self-organizing maps. Each circle on the map represents a set of regions with very similar modification and expression profiles; neighbouring circles on the map are also similar. Black line separates H3K9me2-enriched regions. Scale is in relative enrichment calculated by centring input-normalised RPKM (for ChIP), expression and DNA methylation and dividing by standard deviation. (**D**) Schematic of early mouse development showing numbers of genes becoming activated (red) and repressed (green) in E6.25 when compared with E3.5 ICM (Log2(RPKM)<4, p-value<0.05, Log2(FC) >1). (**E**) Pie charts showing proportion of genes repressed or activated in E6.25 with enrichment of H3K9me2 or H3K27me3 at promoter and/or gene body. Histone modification enrichment was classified using k-means. (**F**) Bar plots show fold enrichment of selected GO terms in genes repressed and activated in E6.25 epiblast compared with E3.5 ICM. * p-value<0.05 using Fisher test. (**G**) GO term enrichment analysis of genes repressed in E6.25 epiblast when compared with E3.5 ICM in relation to their histone methylation status. X axis shows the GO term enrichment bias between H3K9me2- and H3K27me3-marked genes (GO term fold enrichment in H3K9me2 marked genes vs.

*Figure 3 continued on next page*

*Figure 3 continued*

that in H3K27me3 enriched ones). Y axis is minimum Fisher p-value in the H3K9me2- or H3K27me3-marked genes. Complexity of enriched GO terms was reduced by removing terms with highly overlapping gene sets. Also see *Figure 3—figure supplement 1–3*. lcChIP-seq: low cell number chromatin immunoprecipitation with sequencing; H3K9me2: histone H3 lysine 9 dimethylation; H3K27me3: histone H3 lysine 27 trimethylation; RPKM: Reads Per Kilobase of transcript per Million mapped reads; FC: fold change; ICM: inner cell mass; GO: gene ontology.

The following figure supplements are available for figure 3:

**Figure supplement 1.** LcChIP-seq on E6.25 epiblast.

**Figure supplement 2.** H3K9me2 and H3K27me3 correlate with distinct CpG and DNA methylation states.

**Figure supplement 3.** Accumulation of H3K9me2 or H3K27me3 at promoters of genes becoming repressed depends on CpG content.

in EpiSCs (*Figure 4—figure supplement 1B*). This observation indicates that EpiSC derivation leads to an aberrant epigenetic state, especially at germline promoters. Such stable silencing of these genes might contribute to a reduced competence of EpiSC for PGC fate.

## G9a and EZH2 directly regulate specific sets of genes

Next, we integrated our results on the transcriptional and epigenetic state of epiblast cells from embryos in vivo to identify genes under direct epigenetic regulation by G9a and EZH2. We found that >97% of genes enriched with either H3K9me2 or H3K27me3 modifications do not exhibit differential expression in the $Ehmt2^{-/-}$ and $Ezh2^{-/-}$ embryos (*Figure 5A*). This is consistent with only a minority of Polycomb Repressive Complex 2 (PRC2) targets being dependent on H3K27me3 for their repression in naïve ESCs (*Riising et al., 2014*). Thus, the epigenetic programming of epiblast cells by G9a and EZH2 appears to directly regulate silencing at only a limited set of genes, possibly owing to redundant epigenetic mechanisms. Nevertheless, we found that ~63% of genes upregulated in $Ezh2^{-/-}$ and ~36% in $Ehmt2^{-/-}$ show enrichment for H3K27me3 and H3K9me2, respectively (*Figure 5A*, *Figure 5—figure supplement 1A*, *Figure 5—source data 1,2*). In the case of $Ehmt2^{-/-}$, a larger proportion (~53%) of genes seem to be regulated by H3K9me2 deposition at the gene bodies (*Figure 5A*). Among direct G9a targets in the epiblast are *Asz1*, *Casp14* and *Cdkn1a*, but not *Otx2*, or polycomb targets such as *Hoxc10* (*Figure 5B*). These overlaps are significantly higher than the 10% previously observed when analysing *Suz12* knockout (KO) sESC, which self-renew without PRC2 and accumulate secondary transcriptional alterations (*Pasini et al., 2007*). Thus, by focusing on in vivo epiblast just prior to the onset of an overt phenotype, we were able to identify direct primary targets of both G9a and EZH2.

Due to very limited material, we confirmed our genome-wide analysis of epiblast by using a more tractable in vitro model of priming cells for gastrulation. The in vivo developmental progression is represented in vitro by 2i/LIF ESCs, which are equivalent to ICM, followed by primed EpiLCs and EpiSCs, which represent postimplantation epiblast and the primitive streak, respectively (*Figure 1A*) (*Hayashi et al., 2011*; *Kojima et al., 2014*). The lcChIP-qPCR analysis showed rapid increase in H3K9me2 levels in primed cells at direct targets of G9a, including *Asz1* promoter and *Cdkn1a* gene body (*Figure 5C*), as well as promoters of other germline and proliferation regulators (*Figure 5—figure supplement 1B*). Thus, our genome-wide dataset together with the analysis of single *loci* reveal that G9a represses regulators of germline and cell proliferation, and a subset of them accumulate H3K9me2 mark shortly after implantation.

## Role of G9a at transposable elements

We next investigated the role of G9a in regulating transposable elements (TEs) as previous studies have shown that loss of G9a in sESC leads to aberrant expression of murine endogenous retroviruses with leucine tRNA primer (MERV-L) (*Maksakova et al., 2013*). To this end, we combined our lcChIP-seq and RNA-seq from E6.25 epiblast and mapped only unique reads to all mouse repeat loci with sufficient coverage. We found that ~15% ($4.8 \times 10^5/3.3 \times 10^6$) and ~18% ($4.8 \times 10^5/2.6 \times 10^6$) of repeats show significant H3K9me2 and H3K27me3 enrichment, respectively (*Figure 5—figure supplement 2A*). On the other hand, only ~0.11% ($3636/3.3 \times 10^6$) and ~0.06% ($1646/2.6 \times 10^6$) of all

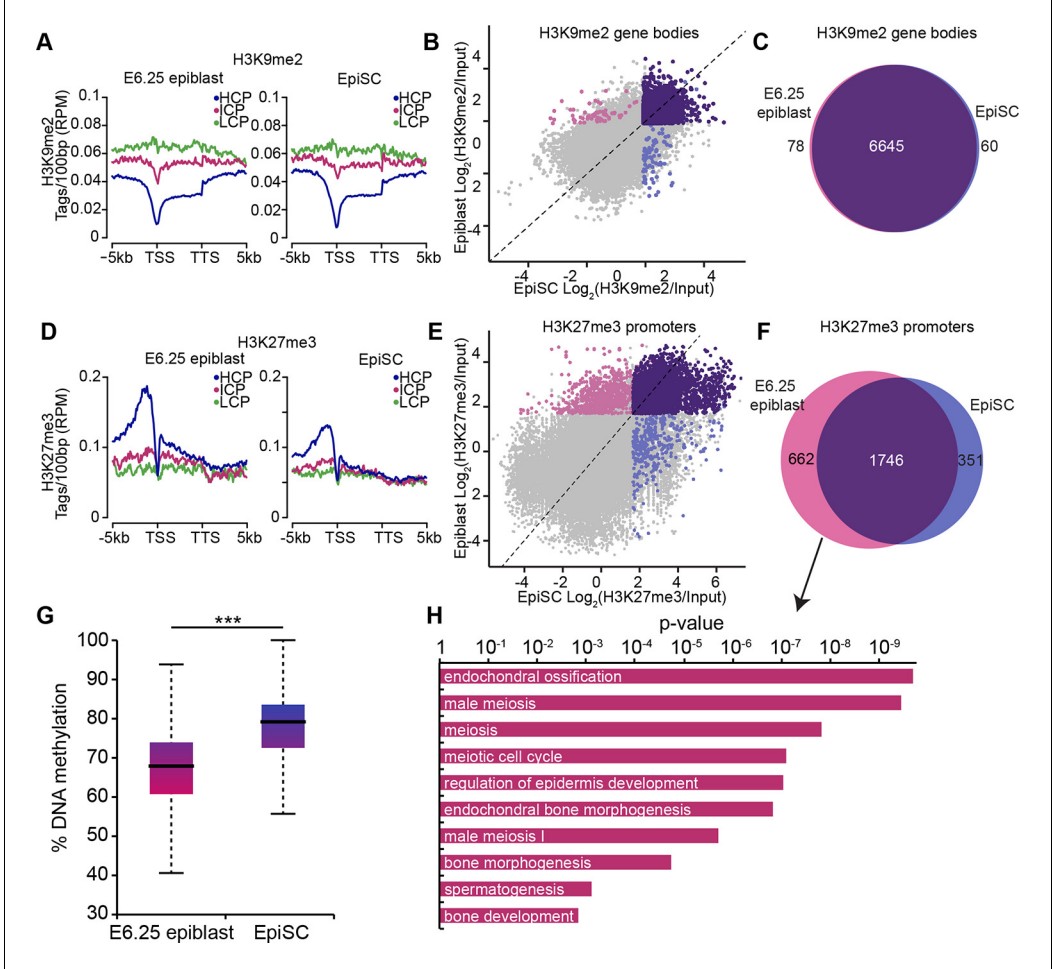

**Figure 4.** EpiSC show aberrant H3K27me3 distribution and DNA hypermethylation at germline genes. (A,D) Distribution of H3K9me2 (A) and H3K27me3 (D) by metagene analysis in epiblast and EpiSCs. Genes were classified based on promoter CpG density. (Scale = base pairs.) (B,E) Scatter plots showing H3K9me2 (C) and H3K27me3 (D) enrichment at gene bodies and promoters, respectively, in EpiSC versus E6.25 epiblast. Regions were classified as marked in both samples (purple), epiblast (pink) or EpiSC only (blue). Classification was performed using k-means and EdgeR (pval<0.05, Log2(FC)>2). (C,F) Venn diagrams showing overlap of H3K9me2 (E) or H3K27me3 (F) enrichment between EpiSC and E6.25 epiblast. (G) Box plot showing global DNA methylation levels from WGBSeq in E6.5 epiblast and EpiSC. ***p-value<0.001 from Wilcoxon rank sum test. (H) Bar plot showing top 10 enriched GO terms in genes uniquely marked by H3K27me3 in the E6.25 epiblast. Also see *Figure 4—figure supplement 1*. HCP: high CpG density; ICP: intermediate CpG density; LCP: low CpG density; TSS: transcriptional start site; TTS: transcriptional termination site; EpiSCs: epiblast stem cells; H3K27me3: histone H3 lysine 27 trimethylation; H3K9me2: histone H3 lysine 9 dimethylation; FC: fold change; GO: gene ontology; WGBSeq: whole genome bisulfite sequencing.

The following figure supplement is available for figure 4:

**Figure supplement 1.** H3K9me2, H3K27me3 and DNA methylation dynamics between E6.25 epiblast and EpiSC.

repeats were significantly upregulated in *Ehmt2*[−/−] and *Ezh2*[−/−] epiblast, respectively. By intersecting fractions of upregulated and epigenetically marked loci in each subfamily, we identified repeats that are enriched for and regulated by H3K9me2 or H3K27me3 (*Figure 5D*, *Figure 5—figure supplement 2B–F*). Direct targets for EZH2 are limited and exhibited no subfamily trend. However, 13/15 subfamilies repressed by G9a are notable as they correspond to endogenous retroviral elements (ERV), especially ERV-L and ERV-LMaLR (*Figure 5—figure supplement 2C*). Consistent with the RNA-seq analysis, single cell RT-qPCR from individual E6.25 *Ehmt2*[+/+] or *Ehmt2*[−/−] epiblast cells showed upregulation of mouse transposon D (MTD), mouse transposable element b (MTEb ) and Lx8 but not of IAP repeat elements (*Figure 5E*). To confirm G9a-dependent deposition of H3K9me2

at these transposons, we used an in vitro model. We generated two *Ehmt2^{F/−}* ESC lines expressing a tamoxifen (TAM)-inducible Cre recombinase (CreER), which were cultured in 2i/LIF medium. Following TAM treatment, day 2 EpiLCs were used for lcChIP-qPCR, which showed significant depletion of H3K9me2 upon loss of G9a at MTEb and Lx8 loci, but not IAP (*Figure 5F*). Thus, G9a deposits H3K9me2 at some repeat elements in postimplantation epiblast, where it is necessary to repress TEs that predominantly belong to the ERVs class. This implies a potentially important role for G9a in maintaining genomic integrity.

## H3K9me2 modification encompasses enhancers undergoing inactivation in primed pluripotent cells

H3K9me2 modification, unlike H3K27me3, accumulates rapidly in the postimplantation epiblast (*Figure 1B*), preferentially in the intergenic regions where many *cis* regulatory elements reside (*Figure 4—figure supplement 1A*). Visual inspection of our lcChIP-seq analysis revealed that multiple putative enhancers show H3K9me2 enrichment in the epiblast, for example, in close proximity to *Esrrb* and *Prdm1* (*Figure 6A*), but the role of this modification at such elements is thus far unknown.

To examine likely functions of H3K9me2 at enhancers, we turned to the in vitro model of ESC priming towards EpiLCs and EpiSCs. We generated high quality native ChIP-seq datasets for H3K27me3 and H3K9me2 from 2i/LIF ESCs, EpiLCs and EpiSCs. Enhancers in these pluripotent cells have recently been identified based on enrichment of p300, H3K4me1 and H3K27ac (*GSE56138*, *GSE57409*) (*Buecker et al., 2014*; *Factor et al., 2014*). Consistent with programming of epiblast in vivo by H3K9me2 modification (*Figure 1B*), we found genome-wide accumulation of H3K9me2 in primed pluripotent cells (*Figure 6—figure supplement 1A*). As H3K9me2 ChIP-seq measures only the relative enrichment, we have scaled it to indicate absolute quantities of histone modification. Our analysis revealed increased enrichment of H3K9me2 at genes upon exit from naïve pluripotency (*Figure 6—figure supplement 1B*). This effect extends to many enhancers but significant H3K9me2 enrichment is observed at elements, which show DNA hypermethylation and cluster separately from poised H3K27me3-enriched elements (*Figure 6—figure supplement 1D,E*, *Figure 6—figure supplement 2A,B*). As with H3K27me3 modification, H3K9me2 is not a distinct enhancer mark, but rather is associated with larger domains within which regulatory elements reside (*Figure 6—figure supplement 2C*). Interestingly, of all the enhancers that are active in 2i/LIF ESCs (p300, H3K4me1 and H3K27ac enrichment), ~12% (3884) gain H3K9me2 after 2 days of induction towards EpiLCs (*Figure 6B*), while only 3% (1117) become enriched for H3K27me3. Such elements are also enriched for H3K9me2 in E6.25 epiblast (*Figure 6B*). To validate these findings, we performed lcChIP-qPCR from ethanol or TAM treated *Ehmt2^{F/−}* EpiLCs expressing CreER. We confirmed that previously identified enhancer elements gain H3K9me2 in a G9a-dependent manner but active or H3K27me3 poised elements do not (*Figure 6C*). Thus, acquisition of H3K9me2 occurs at distal regulatory elements, perhaps to direct or reinforce their inactivation.

Notably, H3K9me2-marked enhancers retain partial enrichment of active H3K27ac mark in EpiLCs, but exhibit near-complete H3K27ac loss in EpiSCs (*Figure 7A*, *Figure 6—figure supplement 1C*). Consistently, enhancer classification based on their epigenetic state has shown that ~57% (2212) of H3K9me2-enriched enhancers in EpiLCs still retain significant H3K27ac (*Figure 7—figure supplement 1*), which we further confirmed using lcChIP-qPCR (*Figure 7B*). To exclude possible effects of cell population heterogeneity, we performed sequential ChIP-qPCR, which showed dual H3K9me2 and H3K27ac enrichment (*Figure 7C*). Such transient dual marking might indicate their continued responsiveness to signalling cues. In addition, H3K9me2-enriched elements showed no change in the active H3K27ac modification in EpiLCs lacking G9a (*Figure 7D*). Taken together, our findings show that the repressive H3K9me2 and activating H3K27ac modifications transiently co-occur at enhancers during epiblast priming.

## G9a mediates efficient inactivation of enhancers

Enrichment of H3K9me2 generally coincides with transcriptional repression in epiblast cells that prompted us to incorporate our RNA-seq and published data from microarray experiments (GSE30056) (*Hayashi et al., 2011*). We found that enhancers marked by H3K9me2 are in close proximity to repressed genes, both in the epiblast in vivo and EpiLCs in vitro (*Figure 8A,B*). Consistent with the *Ehmt2^{−/−}* embryonic phenotype affecting growth and development, these genes are

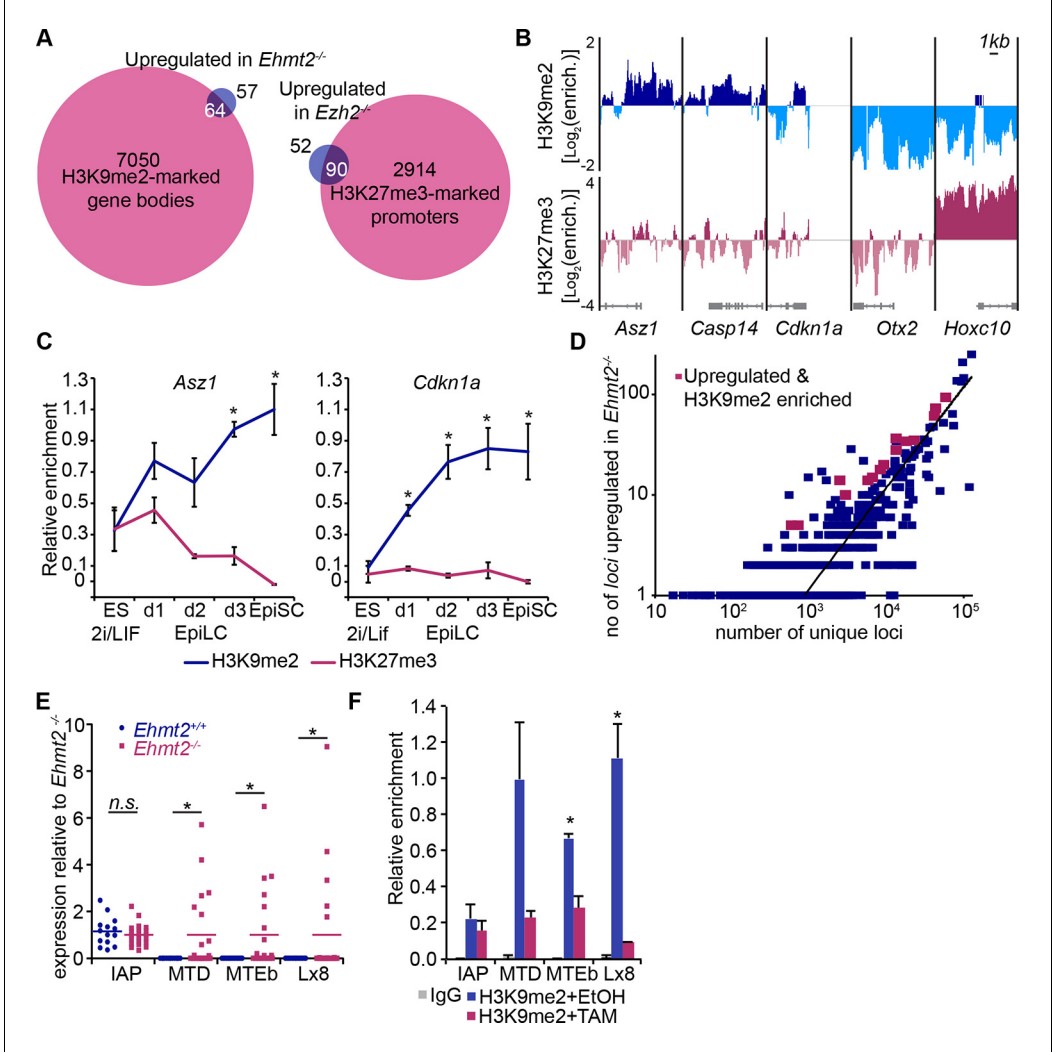

**Figure 5.** H3K9me2 and H3K27me3 are directly involved in repression of genes and transposable elements. (**A**) Venn diagrams showing overlap between H3K27me3 enrichment at promoters (left panel) or H3K9me2 at gene bodies (right panel); genes upregulated in $Ezh2^{-/-}$ and $Ehmt2^{-/-}$ epiblasts are shown. The overlaps are statistically significant (p-value< 0.01 using $Chi^2$ test). (**B**) Genome browser tracks showing H3K9me2 and H3K27me3 enrichment at genes that are: derepressed in $Ehmt2^{-/-}$ (*Asz1, Casp14, Cdkn1a*), active epiblast markers (*Otx2*), and PRC2 targets (*Hoxc10*). Data is shown as a sliding window (1kb and 300bp for H3K9me2 and H3K27me3, respectively) of enrichment over input: Log2(RPM ChIP/RPM Input). (**C**) lcChIP-qPCR validation for H3K9me2 and H3K27me3 at the promoter of *Asz1* and gene body of *Cdkn1a* during EpiLC and EpiSC induction. Signal was scaled relative to average enrichment on negative (*Gapdh*: H3K9me2 and H3K27me3) and positive control regions (*Pcsk5*: H3K9me2, *Hoxc10*: H3K27me3). Data are represented as mean (± SEM) from three independent biological replicates (*Student's t-test p-value<0.05 relative to 2i/LIF ESC sample). (**D**) Scatter plot showing correlation between the number of unique repeat loci in each subfamily with the number of loci upregulated in $Ehmt2^{-/-}$ E6.25 epiblast. Red points are subfamilies with significant H3K9me2 enrichment and increased proportion of upregulated loci. (**E**) Single cell RT-qPCR validation of repeat upregulation in individual E6.25 $Ehmt2^{+/+}$ and $Ehmt2^{-/-}$ epiblast cells. Statistical significance was calculated using Wilcoxon rank sum test for IAP where the majority of WT and KO cells show detectable expression. For remaining repeats, a $Chi^2$ test was used.(*p-value<0.05). (**F**) LcChIP-qPCR measuring H3K9me2 levels at selected repeat elements in $Ehmt2^{F/-}$ $CreER^{+ve}$ d2 EpiLCs treated with EtOH or TAM. Data are mean (± SD) from two independent biological replicates. (*Student's t-test p-value<0.05 of EtOH compared with TAM treated sample). Also see *Figure 5—figure supplement 1,2* and *Figure 5—source data 1,2*. H3K9me2: histone H3 lysine 9 dimethylation; H3K27me3: histone H3 lysine 27 trimethylation; EZH2: Enhancer of zeste homolog 2; ChIP: chromatin immunoprecipitation; EpiLCs: epiblast-like cells; EpiSCs: epiblast stem cells; SEM: standard error of the mean; TAM: tamoxifen; EtOH: ethanol; IAP: intracisternal A particle.

The following source data and figure supplements are available for figure 5:

**Source data 1.** List of promoters enriched for H3K27me3 in lcChIP-seq from E6.25 epiblast data is based on two biological replicates of lcChIP-seq.

**Source data 2.** List of gene bodies enriched for H3K9me2 in lcChIP-seq from E6.25 epiblast Data is based on two biological replicates of lcChIP-seq.

*Figure 5 continued on next page*

*Figure 5 continued*

**Figure supplement 1.** H3K9me2 accumulates at G9a-regulated promoters in the epiblast.

**Figure supplement 2.** H3K9me2 accumulates at G9a-regulated repeat elements in the epiblast.

enriched for the regulators of apoptosis (*Figure 8—figure supplement 1A*). To further address potential functions of these enhancer elements, we performed de novo motif analysis. H3K9me2-marked enhancers showed significant enrichment for regulators of early development (T and SOX2) and apoptosis (p53 and p63) (*Figure 8—figure supplement 1B*). In line with the PRC2 function, H3K27me3 poised enhancers showed enrichment for motifs of skeletal development regulators SOX9 and HOXD8 (*Figure 8—figure supplement 1C*). Such non-overlapping motif signatures further validate that H3K27me3 and H3K9me2 accumulate at distinct enhancers that regulate early development.

Transcriptional analysis of genes most proximate to specific enhancers is subject to errors, since these elements can regulate distant promoters (*Sanyal et al., 2012*). Therefore, we determined the correlation between H3K9me2 enrichment and enhancer activity by analysing the expression levels of enhancer RNAs (eRNA), which is a hallmark of active enhancers (*Kim et al., 2010*). We performed RT-qPCR for candidate robustly expressed eRNAs originating from regions enriched for H3K9me2 or H3K27me3. Upon EpiLC induction, such elements underwent robust transcriptional inactivation (*Figure 8C,D*). This is not a global effect because enhancers for genes, such as *Otx2* that are linked with primed pluripotency, show increased eRNA expression in EpiLCs (*Figure 8D*). Thus, upon blastocyst implantation, H3K9me2 domains extend to active enhancers targeted for initiation of silencing, which accounts for the co-enrichment with H3K27ac.

To determine the functional relevance of H3K9me2 on enhancers, we used two $Ehmt2^{F/-}$ ESC lines expressing CreER. Following TAM treatment, day 2 EpiLCs showed increased eRNA expression at 4 of 8 previously identified H3K9me2-marked enhancers, confirming that, in principle, this histone modification promotes repression of enhancer activity (*Figure 8E*). This was specific to H3K9me2, since the H3K27me3-enriched and active regulatory elements were unaffected (*Figure 8E*). Finally, we sought to validate our findings in vivo since multiple enhancer elements showed increased levels of H3K9me2 in the post-implantation epiblast. To this end, we isolated RNA from individual $Ehmt2^{+/+}$ and $Ehmt2^{-/-}$ E6.25 epiblasts. After complementary DNA (cDNA) preamplification, we measured eRNA expression in these samples. We were able to detect 7/8 eRNAs associated with H3K9me2 marked enhancers, and 4 of these showed significant increase in expression in $Ehmt2^{-/-}$ embryos (*Figure 8F*). Of all four control eRNAs detected in vivo, none showed altered expression. Taken together, these results reveal that G9a contributes to transcriptional and epigenetic repression of a subset of enhancers. These elements are typically active in naïve pluripotent cells but become repressed during priming of the epiblast.

## Discussion

We present here evidence that global H3K9me2 is established by a wave of G9a activity during early postimplantation development and contributes to establishing a crucial chromatin signature at promoters and gene bodies. Once acquired, H3K9me2 represses a specific subset of genes, including key regulators of proliferation and germline development. The enrichment of H3K9me2 additionally extends into domains that contain multiple enhancer elements, leading to their developmentally-linked inactivation. This uncovers an important role for G9a in setting the regulatory circuitry in the epiblast that enables the subsequent developmental programme to unfold.

A wave of heterochromatization occurs during early postimplantation development that contributes to the establishment of a specific 'primed' epigenetic state in epiblast cells (*Borgel et al., 2010*; *Gilbert et al., 2010*). While both G9a and PRC2 are involved in this epigenetic programming process, we found limited overlap between H3K9me2 and H3K27me3 targets or, indeed, between genes that were upregulated in $Ezh2^{-/-}$ and $Ehmt2^{-/-}$ mutant embryos. This indicates that they likely have independent functions, as observed in sESC, which is unrelated to the previously reported

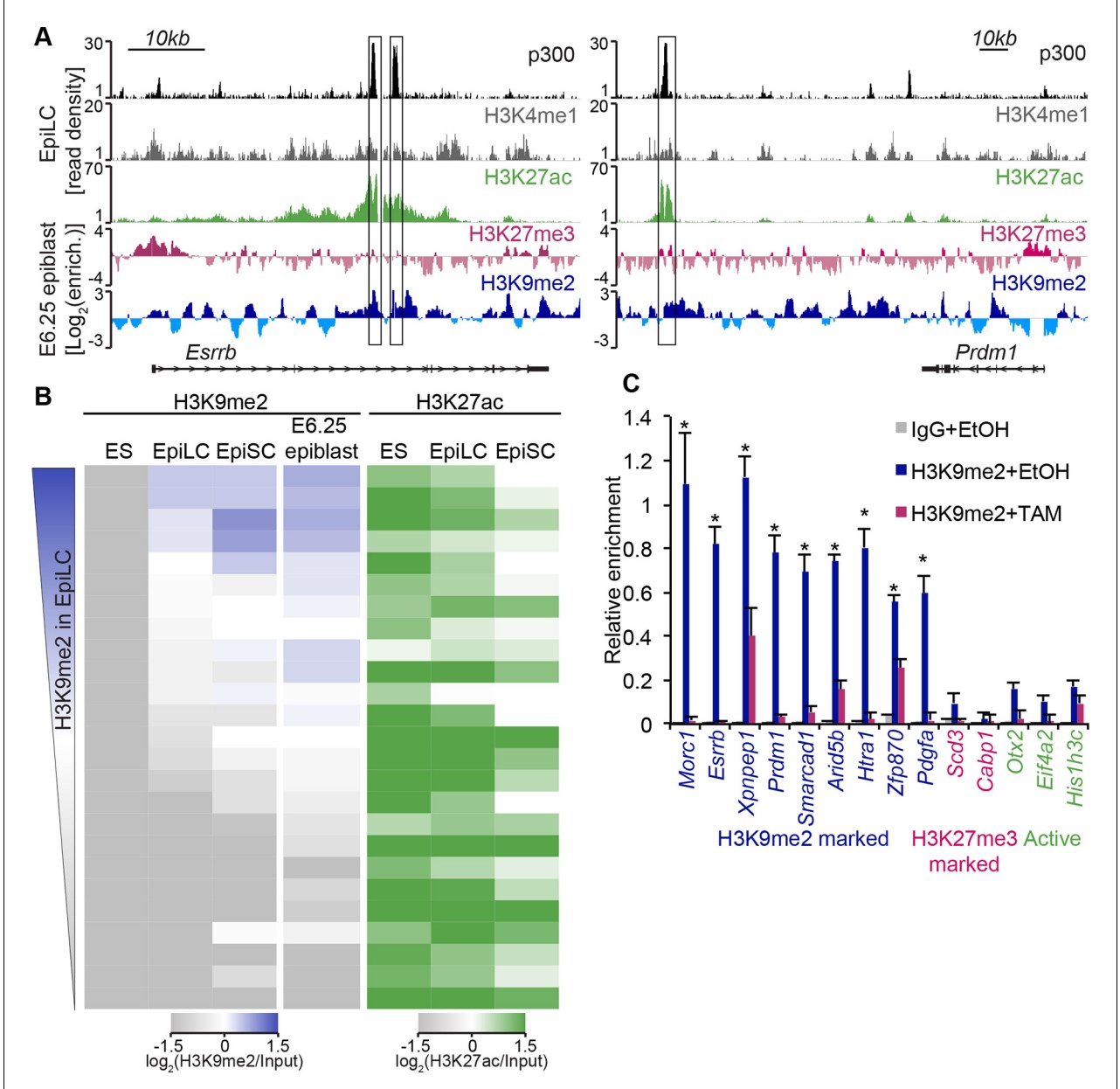

**Figure 6.** H3K9me2 marks enhancers undergoing inactivation during exit from naïve pluripotency. (A) Genome browser tracks showing p300, H3K27ac, and H3K4me1 at inactive *Esrrb* and *Prdm1* putative enhancers (black boxes) in day 2 EpiLCs (*GSE56138*) (*Buecker et al., 2014*). Bottom two tracks show H3K27me3 (red) and H3K9me2 (blue) enrichment in the E6.25 epiblast. Green and grey tracks show read density, while blue and red tracks show Log2 enrichment of ChIP sample over the input sample. (B) Heatmaps showing H3K9me2 and H3K27ac enrichment at enhancers active in ESCs grown in 2i/ LIF. All ESC enhancers in 2i/LIF enriched for p300, H3K27ac and H3K4me1 were clustered using kohonen package based on H3K9me2 and H3K27ac. Individual classes were ranked based on H3K9me2 levels in EpiLCs. H3K27ac tracks were extracted from (*GSE56138, GSE57409*) (*Buecker et al., 2014*; *Factor et al., 2014*). (C) LcChIP-qPCR measuring H3K9me2 levels at putative enhancer elements in *Ehmt2^{F/−} CreER^{+ve}* d2 EpiLCs treated with EtOH or TAM. H3K9me2- (blue), H3K27me3- (red) marked, as well as active (green) regulatory elements are shown. Data are mean (± SD) from two independent biological replicates. (*Student's t-test p-value<0.05 of EtOH compared with TAM treated sample). Also see *Figure 6—figure supplement 1,2*. H3K27ac: histone H3 lysine 27 acetylation; EpiLCs: epiblast-like cells; H3K27me3: histone H3 lysine 27 trimethylation; ChIP: chromatin immunoprecipitation; H3K9me2: histone H3 lysine 9 dimethylation; lcChIP-seq: low cell number chromatin immunoprecipitation with sequencing; TAM: tamoxifen; EtOH: ethanol; SD: standard deviation; ESCs: Embryonic stem cells.

The following figure supplements are available for figure 6:

**Figure supplement 1.** Genome-wide accumulation of H3K9me2 extends to multiple enhancer elements.

*Figure 6 continued on next page*

*Figure 6 continued*

**Figure supplement 2.** H3K9me2 marks a distinct set of enhancers.

G9a-dependent recruitment of PRC2 complex (*Lienert et al., 2011*; *Mozzetta et al., 2013*). Thus, prior to gastrulation, H3K9me2 and H3K27me3 generate distinct repressive chromatin states, linked to DNA hyper- and hypomethylation, respectively. These modifications therefore act as complementary systems to target specific regulatory pathways, ensuring that they become repressed during the exit from naïve pluripotency. Following PGC specification, however, there is germline-specific resetting of the epigenome, including the erasure of H3K9me2 and DNA methylation (*Hajkova et al., 2002*; *Seki et al., 2005*; *2007*), which allows for the expression of germline-specific genes that have been silenced by these epigenetic modifications.

We have characterised the epigenetic landscape of the in vivo primed pluripotent cells relative to its in vitro model represented by EpiSCs. It is noteworthy that the two cell types show highly similar distribution of H3K9me2, but EpiSCs globally gain DNA methylation and lose H3K27me3 from germline-related genes, which might contribute to the reduced competence of EpiSCs towards PGCs. Indeed, stable promoter DNA methylation has been previously reported for two germline genes, *Stella* and *Rex1* (*Bao et al., 2009*; *Hayashi and Surani, 2009*). Thus we show that in vitro derivation and self-renewal promotes aberrant accumulation of epigenetic modifications, as is the case with sESC and hematopoietic stem cells (*Ludwig et al., 2014*; *Weidner et al., 2013*).

Our study on a transient and highly dynamic state of the pregastrulation epiblast reveals an unexpected level of epigenetic regulation. It is established that enhancer elements acquire H3K4me1 and H3K27ac coincident with their developmental activation at this point, while others become poised and enriched for H3K27me3 (*Buecker et al., 2014*; *Factor et al., 2014*). We additionally reveal that G9a is involved in the rapid switching of epigenetic states at regulatory elements by depositing H3K9me2 (*Figure 9*). The accumulation of H3K9me2 at enhancers is mostly linked with transcriptional repression, although they transiently retain significant levels of H3K27ac. Thus, we show uncoupling of histone acetylation from the transcriptional state, which indicates that loss of an active epigenetic signature is often secondary to enhancer inactivation. Such contrasting epigenetic marks at enhancers might confer responsiveness and plasticity of cells to signalling cues. This coincides with the rapidly unfolding developmental programme in postimplantation epiblast, resulting in potential for diverse cell fate decisions. In line with this role, the loss of H3K9me2, as seen at an oestrogen-induced enhancer in breast cancer cells, promotes reactivation of the apoptosis regulator *bcl2* (*Perillo et al., 2008*).

The regulation of enhancer elements likely contributes to the phenotype of G9a null embryos, but is unrelated to increased expression of pluripotency genes, contrary to a previous suggestion (*Yamamizu et al., 2012*). Decreased proliferation and increased apoptosis is more likely due to derepression of negative regulators of cell cycle, including a potent cyclin-dependent kinase inhibitor, *Cdkn1a* (also called p21). Furthermore, G9a preferentially represses three clusters of X-linked genes: *Rhox*, *Xlr* and *Mage-a*. Two of these are imprinted, suggesting a role for H3K9me2 in this process, which merits further investigation (*Maclean et al., 2011*; *Raefski and O'Neill, 2005*).

Pluripotency is a transient state during mammalian development that is established at the blastocyst stage, and undergoes changes after implantation and development of the epiblast. Importantly, G9a-dependent programming entails spreading of H3K9me2 modification to enhancer elements where it regulates their activity; this contributes to rapid and dynamic changes at a critical period of epiblast development prior to gastrulation.

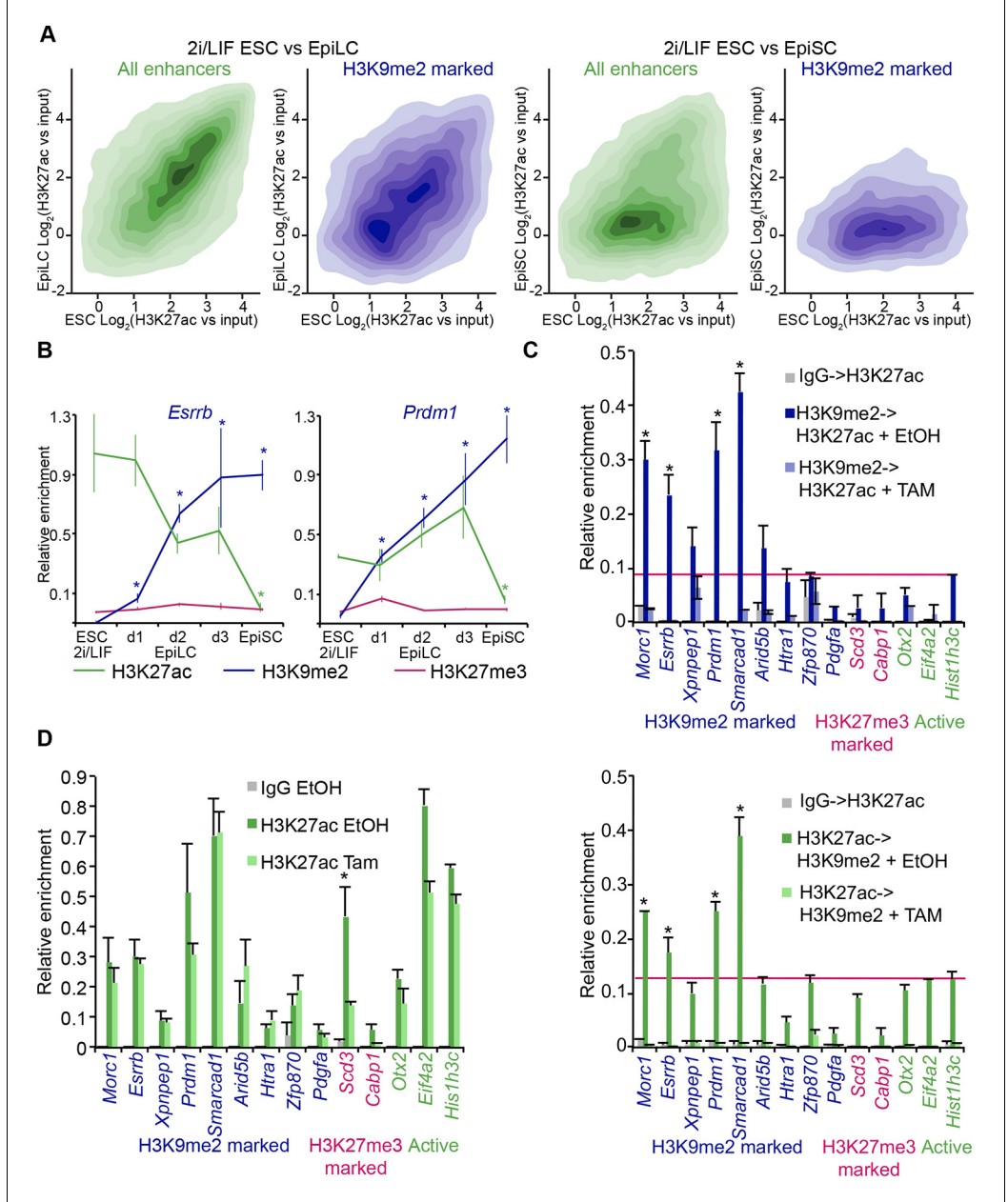

**Figure 7.** H3K9me2 spreading to enhancers results in transient coenrichment with H3K27ac. (A) Density contour plots showing correlation between H3K27ac enrichments at enhancers in 2i/LIF ESCs versus EpiLC (left panels) or 2i/LIF ESCs versus EpiSC (right panels). Green panels show all regulatory elements active in 2i/LIF ESCs (p300, H3K4me2 and H3K27ac enrichment), while blue panels show only the subset that becomes enriched with H3K9me2 in EpiLCs. (B) lcChIP-qPCR results measuring levels of H3K9me2, H3K27me3 and H3K27ac at putative *Prdm1* and *Esrrb* enhancers. Samples were collected during EpiLC and EpiSC induction. Signal was scaled relative to average enrichment on negative (*Pcsk5*: H3K27ac, *Gapdh*: H3K9me2 and H3K27me3) and positive control regions (*Pcsk5*: H3K9me2, *Gpr20*: H3K27ac, *Hoxc10*: H3K27me3). Data are represented as mean ( ± SEM) from three independent biological replicates (*Student's t-test p-value<0.05 relative to 2i/LIF ESC sample). (C) Sequential ChIP-qPCR performed *Ehmt2^F–^ CreER^+ve^* d2 EpiLCs treated with EtOH or TAM. Upper panel shows samples precipitated first with anti-H3K9me2 antibody and later H3K27ac. Bottom panel shows results from an inverse experiment. Samples were scaled to a positive (*Gpr20*) control region. Data are mean (± SD) from two independent biological replicates. (*Student's t-test p-value<0.05) (D) LcChIP-qPCR measuring enrichment of H3K27ac at selected enhancers in EtOH- or TAM-treated *Ehmt2^F–^ CreER^+ve^* d2 EpiLCs. H3K9me2-marked (blue) and control H3K27me3-poised or active regions are presented. Data are mean (± SD) from two independent biological replicates. (*Student's t-test p-value<0.05). Also see *Figure 7—figure supplement 1*. H3K27ac: histone H3 lysine 27 acetylation; 2i/LIF: two-inhibitor/leukemia inhibitory factor; EpiLCs: epiblast-like cells; EpiSCs: epiblast stem cells; lcChIP-seq: low cell number chromatin immunoprecipitation with sequencing; H3K9me2: histone H3 lysine 9 dimethylation; H3K27me3: histone H3 lysine 27 trimethylation; SEM: standard error of the mean; ESCs: Embryonic stem cells; TAM: tamoxifen; EtOH: ethanol; SD: standard deviation.

*Figure 7 continued on next page*

*Figure 7 continued*

The following figure supplement is available for figure 7:

**Figure supplement 1.** H3K9me2-marked enhancers retain some H3K27ac in EpiLCs.

## Materials and methods

### Experimental procedures

#### Mouse breading, embryo collection, and staging

Timed natural matings were used for all experiments. Noon of the day when the vaginal plugs of mated females were identified was scored as E0.5. When necessary, postimplantation embryos were staged as previously described (*Downs and Davies, 1993*).

For *Ehmt2* matings, a published conditional allele was used (*Sampath et al., 2007*). *Ehmt2*$^{+/-}$ mice were next crossed with a Δ*PE-Pou5f1-EGFP* reporter line (GGOF) (*Yeom et al., 1996*). This transgene drives enhanced green fluorescent protein (EGFP) expression in preimplantation ICM, and residual fluorescence persists in the epiblast until ~E6.75. For the collection of EZH2-deficient embryos, a previously described allele was used (*O'Carroll et al., 2001*). All husbandry and experiments involving mice were carried out according to the local ethics committee and were performed in a facility designated by the Home Office.

#### ESC derivation

Two male *Ehmt2*$^{F/-}$ *GGOF*$^{+ve}$ ESC lines were derived in 2i/LIF conditions as previously described (*Nichols et al., 2009*). Cells were cultured in N2B27 2i/LIF conditions on gelatine supplemented with 1% knockout serum replacement (*Ying et al., 2008*). To generate inducible *Ehmt2* KO lines, *Ehmt2*$^{F/-}$ *GGOF*$^{+ve}$ ESCs were transfected with an expression vector for TAM-inducible CreER by the use of lipofectamine 2000 (Life Technologies, CA). Cre recombination was induced by the addition of TAM. After a 2-day treatment, no Flox G9a allele was detected by genotyping.

#### Epiblast isolation

For ChIP experiments, epiblast cells were isolated from E6.25 pre-gastrulating embryos coming from outbred MF1 females crossed with GGOF stud males. After recovering embryos from the decidua, Reichert's membrane was dissected out and extraembryonic cone was also removed. Remaining tissue was used to prepare single cell suspension for sorting using MoFlo high-speed cell sorter (Beckman Coulter, CA) based on EGFP expression. The purity of epiblast cells was assessed by staining for SOX2 and was in excess of 95%. For single epiblast experiments, embryos were dissected as previously described (*Tesar et al., 2007*).

#### RNA isolation, reverse transcription and qPCR

Total RNA was extracted using PicoPure RNA isolation Kit (Life Technologies, CA) or AllPrep DNA/RNA Micro Kit (Qiagen, Germany) with an on-column DNaseI digestion (Qiagen, Germany). cDNA was prepared using SuperScript III (Life Technologies, CA) and random hexamer primers (Life Technologies, CA). RT-q PCR reactions were performed using Kapa Sybr Fast qPCR kit (Kapa Biosystems, MA). Single-cell RT-qPCR was performed as previously described, for detailed protocol and reagents refer to the original publication (*Tang et al., 2010*). For primer sequences see *Supplementary file 1*. Amplification was performed with QuantStudio 6 Flex Real-Time PCR system (Life Technologies, CA).

#### Single epiblast RNA-seq

Dissected *Ehmt2*$^{+/-}$× *Ehmt2*$^{+/-}$ or *Ezh2*$^{+/-}$× *Ezh2*$^{+/-}$E6.25 epiblasts were lysed and RNA was extracted like for RT-qPCR. Quality and concentration of eluted RNA was assessed with the Agilent RNA 6000 Pico Kit (Agilent Technologies, CA). Only samples with the RNA integrity score >8 were further processed. For *Ezh2*, three control and three KO samples from three litters were used. For *Ehmt2*, four controls and four KO from four litters were used. A total of 750 pg of RNA from each

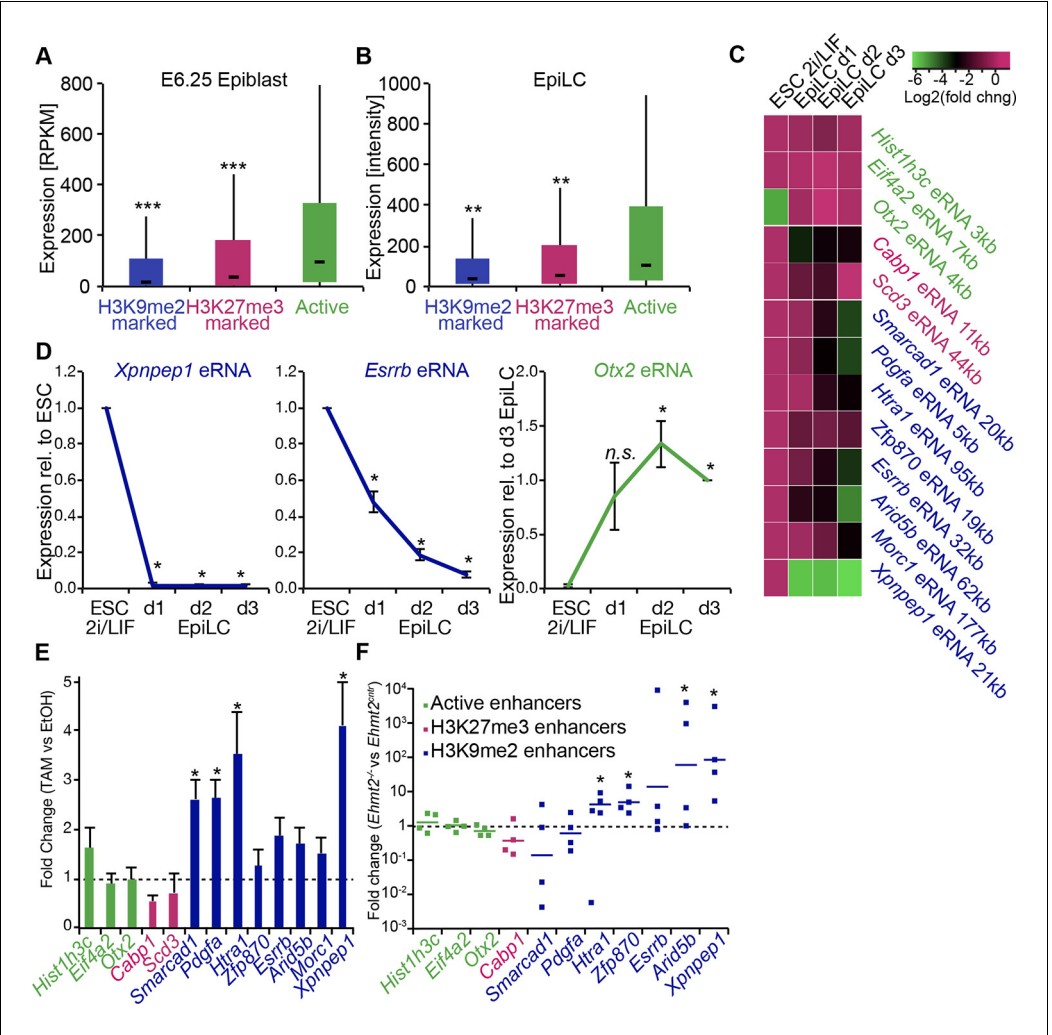

**Figure 8.** G9a promotes transcriptional inactivation of enhancers. (**A,B**) Box plots showing transcript levels of genes in epiblast (**A**) and EpiLCs (**B**), which lie in proximity of H3K9me2- or H3K27me3-marked enhancers. All comparisons are statistically significant (p<0.05). The effect size relative to active set is shown in the graphs: *r≤0.10; **0.10<r≤0.15; ***r>0.15 calculated using Wilcoxon rank sum test. For analysis of EpiLCs, a published microarray experiment was used (GSE30056) (*Hayashi et al., 2011*). (**C,D**) RT-qPCR for eRNAs at selected H3K9me2-marked (blue) and control (red: H3K27me3 enriched, green: active) enhancers shown as line plots (**D**) and summarized in a heatmap (**C**). Data are represented as mean (± SEM) from three independent biological replicates. Samples are normalised to ESC 2i/LIF sample with the exception of *Otx2*, which is normalised to d3 EpiLCs. (*Student's t-test p-value<0.05) (**E**) FC of eRNA expression in *Ehmt2^{F/−} CreER^{+ve}* d2 EpiLCs treated with TAM relative to EtOH control. Transcripts are originating from H3K9me2- (blue), H3K27me3-marked (red) or active (green) *loci*. Data are presented as mean (± SEM) from three independent biological replicates. (*Student's t-test p-value <0.05) (**F**) FC of eRNA expression in individual *Ehmt2^{−/−}* E6.25 epiblasts normalised to *Ehmt2^{+/+}* littermates. Lines show geometric means. (*p<0.05 Wilcoxon rank sum test). Also see *Figure 8—figure supplement 1*. EpiLCs: epiblast-like cells; H3K9me2: histone H3 lysine 9 dimethylation; H3K27me3: histone H3 lysine 27 trimethylation; RT-qPCR: real-time quantitative polymerase chain reaction; eRNA: enhancer RNA; ESCs: Embryonic stem cells; TAM: tamoxifen; EtOH: ethanol.

The following figure supplement is available for figure 8:

**Figure supplement 1.** H3K9me2 enriched enhancers are preferentially linked to the p53 pathway.

sample was amplified using Ovation RNA-seq System V2 (NuGEN Technologies, CA). The quality of cDNA was confirmed by measuring the expression of *Ehmt2*, *Ezh2*, *Pou5f1* and *Nanog* by qPCR.

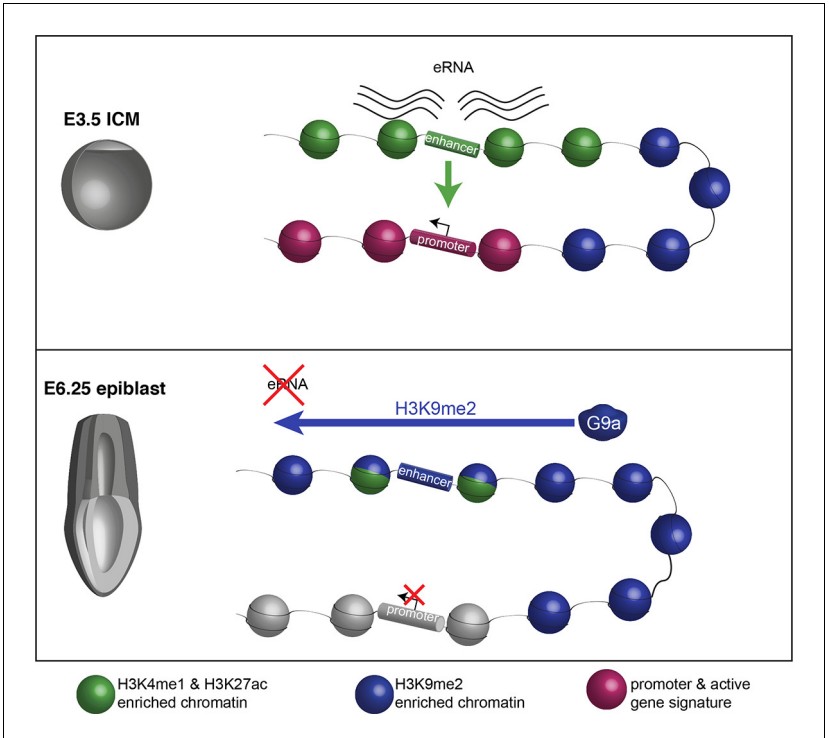

**Figure 9.** Proposed model for G9a-mediated enhancer inactivation. Distal regulatory elements active in ICM are typically associated with H3K4me1 and H3K27ac enrichment (green) as well as eRNA expression. Such elements activate promoters (purple) to increase gene transcription. Following implantation, the majority of enhancers undergo inactivation. In many cases, this process is aided by spreading of G9a-dependent H3K9me2 enrichment domains. Despite retaining H3K27ac and H3K4me1 enrichment, these enhancers typically loose eRNA expression.

For every sample, 1.5 µg of cDNA was sheared to ~230 bp using S220 Focused-ultrasonicator (Covaris, MA). The fragmented cDNA was then concentrated using Qiagen Reaction Cleanup Kit (MinElute). A total of 500 ng of each sample was used as input for library preparation using Encore Rapid DR Multiplex Library System (NuGEN Technologies, CA). Finally, the adaptor-ligated DNA was quantified using KAPA Library Quantification Kit (Kapa Biosystems, MA) and sequenced using HiSeq2000 or HiSeq2500 with single-end 40 or 50 nt read length.

## RNA-seq analysis

RNA-seq reads were all trimmed to 40 nt read length, adapters were removed, and reads were aligned with Tophat2 (*Kim et al., 2013*) against the mouse reference (GRCm38/mm10) genome. Read counts per ENSEMBL transcript were obtained by HTseq-count (*Anders et al., 2014*). Transcript annotations were based on ENSEMBL Release 74. Replicates were evaluated, read counts per transcript were normalised, and analysis of differential expression was performed by using the R Bioconductor DESeq package (*Anders and Huber, 2010*). Normalised read counts were further divided by transcript length (per kB). To account for expression heterogeneity between embryos, the expression in each KO embryo was compared with all control samples. The p-values for each of the three or four comparisons were combined by a Fisher's combined probability test and set at $p<0.05$ (Fisher, 1925). To identify biologically relevant results, we set a minimal expression in each replicate of the upregulated sample at Log2(RPKM)>1. Finally, in each comparison, FC in expression had to be at least Log2(FC) >1.4.

Published RNA-seq datasets for E3.5 blastocysts were downloaded from the European Nucleotide Archive (ERP005749), reads were trimmed to 40 nt length, and subsequently processed in a similar manner for comparison with E6.25 epiblast using DESeq.

## Repeat expression analysis

RepeatMasker annotations for the mouse reference genome were obtained from the Univeristy of California, Santa Cruz (UCSC) Table Browser. RNA-seq reads were aligned to the mouse reference genome by using bowtie (bowtie-bio.sourceforge.net; version 1.1.0) with parameters '-m 1 –v 2 –-best –-strata' in order to select reads that uniquely map to single repeat copies in the genome only. Read counts for repeat regions were normalised by the total number of RNA-seq reads that aligned to protein-coding genes and by repeat size (in kB). Statistical tests for differential expression of genomic repeat copies were performed by the R Bioconductor DESeq package.

## GO term enrichment analysis

GO term enrichment analysis was performed using the DAVID tool (http://david.abcc.ncifcrf.gov, [*Huang et al., 2009*]). P-values presented were calculated using modified Fisher exact p-value. To summarize relative enrichment of GO terms in H3K9me2- versus H3K27me- marked and repressed genes, we have used 'revigo' tool to remove terms with substantial overlapping gene sets (http://revigo.irb.hr, [*Supek et al., 2011*]).

## Immunofluorescence

Embryos were dissected from the decidua and Reichert's membrane and treated as previously described (*Nichols et al., 2009*). Primary antibodies used are as follows: anti-H3K9me2 (Millipore, CA, 17–648), anti-H3K9me2 (Abcam, UK, ab1220), anti-GFP (Nacalai tesque, Japan, GF090R), anti-G9a (R&D Systems, MN, A8620A), anti-GLP (R&D Systems, MN, PP-B0422-00), anti-cleaved Caspase 3 (Abcam, UK, ab32042), anti-Ki67 (BD Bioscience, NJ, 550609), anti-NANOG (Cosmobio, Japan, REC-RCAB002P-F), anti-AP2γ (Santa Cruz, CA, sc-9877). All imaging was performed using SP5 or SP7 confocal microscope (Leica, Germany).

## Stem cell culture

EpiLCs were induced from 2i/LIF ESCs as previously described (*Hayashi et al., 2011*). For genome-wide studies a X6⁻ EpiSCs (*Gillich and Hayashi, 2011*) and GGOF ESC lines were used. EpiSCs were grown in feeder-free conditions in fibronectin (Millipore, CA) coated dishes. Culture media contained: N2B27 medium supplemented with bFGF (12 ng/ml) and ActivinA (20 ng/ml).

## Native ChIP-seq

Chromatin was released as previously described (*Hackett, et al., 2013*). For each ChIP, $10^7$ female day 2 EpiLCs or EpiSCs were used. Immunoprecipitation was performed in dilution buffer (16.7 mM Tris-HCl, pH8, 167 mM NaCl, 1.2 mM ethylenediaminetetraacetic acid [EDTA], 1.1% Triton X-100, 0.01% sodium dodecyl sulfate (SDS), 0.2 mM PMSF, 1 mM dithiothreitol [DTT], 1X Protease Inhibitors) using Protein G Dynabeads (Life Technologies, CA) coated with antibodies specific to H3K27me3 (07–449, Millipore, CA) and H3K9me2 (ab1220, Abcam, UK). For H3K27me3, the ChIP beads were washed three times in ChIP W1 buffer (150 mM NaCl, 10 mM TrisHCl pH 8, 2 mM EDTA, 1% NP40, 1% Na-deoxycholate, 0.2 mM PMSF, 1 mM DTT) and further three washes with increased NP40 concentration to 1.5% (W1.5). For H3K9me2 there were two washes each in low salt (0.1% SDS, 1%, TritonX-100, 2 mM EDTA, 20 mM Tris-HCl, pH 8.1, 150 mM NaCl, 1 mM DTT), high salt (0.1% SDS, 1%, TritonX-100, 2 mM EDTA, 20 mM Tris-HCl, pH 8.1, 300 mM NaCl, 1 mM DTT) and LiCl buffer (0.25 M LiCl, 1% NP40, 1% Na deoxycholate, 1 mM EDTA, 10 mM Tris-HCl pH 8.1, 1 mM DTT). After a final TE (10 mM Tris-HCl pH 8.1, 1 mM EDTA) wash samples were eluted, proteins digested with proteinase K, and DNA purified using phenol/chlorophorm/IAA extraction and further ethanol precipitation. For each experiment, enrichment was confirmed by qPCR for control regions (for primer sequence see *Supplementary file 1*).

Each experiment was performed in biological replicates and 20 ng of immunoprecipitated or input DNA was used for library preparation using Ovation Ultralow DR Multiplex System (0331, NuGEN Technologies, CA). Once prepared, library DNA was resolved on a 2% agarose gel and the mononucleosome (for H3K27me3) and dinucleosome (for H3K9me2) fractions were isolated using MiniElute gel extraction kit (Qiagen, Germany) and sequenced using HiSeq2000 with single-end 40 nt read length.

## Low cell number ChIP-seq

lcChIP-seq was performed by using a modified published method (*Ng, et al., 2013*). Briefly, EpiSC, EpiLCs or FACS-purified (Fluorescence-activated cell sorting) epiblast cells were fixed in 1% formaldehyde (room temperature, 10 min), quenched with 1 vol. 250 mM glycine (room temperature, 5 min), and rinsed with chilled TBSE buffer (20 mM Tris-HCl, 150 mM NaCl, 1 mM EDTA) twice before freezing in liquid nitrogen. After thawing on ice, fixed cells were pooled (25,000 cells per lcChIP-seq and 50,000 cells per lcChIP-qPCR) and lysed with 100 µl 1% SDS lysis buffer (50 mM Tris-HCl pH8, 10 mM EDTA, 1% SDS, Roche protease inhibitor cocktail; 5mM sodium butyrate on ice, 5 min) and then centrifuged (2000 RPM, 10 min). Samples were resuspended in 100 µl of dilution buffer. Samples were sonicated nine times (30 s pulses with 30 s break interval) using a Bioruptor water bath sonicator (Diagenode, Belgium). Chromatin extracts were then pre-cleared with Protein G Dynabeads and immunoprecipitated overnight with Protein G Dynabeads coupled with antibodies specific to H3K27me3 (07–449, Millipore, CA), H3K9me2 (ab1220, Abcam, UK), H3K27ac (ab4729, Abcam, UK) or normal rabbit serum (Santa Cruz, CA). On the next day, beads were washed for 5 min at 4°C once in each low salt, high salt, LiCl and TE buffer. After elution, samples were digested with proteinase K and reverse crosslinked for 6 hr at 68°C. DNA was purified (phenol-chloroform extraction) and used for qPCR analysis to validate enrichments.

Unless otherwise stated, lcChIP-qPCR samples were rescaled by normalizing to +ve control region (H3K9me2: *Pcsk5*; H3K27me3: *Hoxc10*; H3K27ac: *Gpr20*).

For lcChIP-seq, isolated DNA was primed using WGA4 kit (Sigma-Aldrich, MO). The next step involved library amplification using HiFi Library Amplification master mix (Kapa biosystems, MA) and BpmI-primer (CCGGCCCTGGAGTGTTGGGTGTGTTTGG). These reactions were incubated in a thermocycler using following conditions: 98°C for 3 min; 11–12× (98°C for 10 s; 65°C for 30 s; 72°C for 1 min); 72°C for 7 min: or 4°C as suggested. The number of cycles depended on the amount of DNA precipitated and so 11 cycles were used for H3K9me2 and 12 for H3K27me3. Amplified DNA was purified with Agencourt RNA clean XP beads. Adapter trimming was performed by BpmI digestion, secondary adaptor ligation, and a second round of digestion (*Ng, et al., 2013*). Digested DNA was purified with Agencourt RNA clean XP bead and used for library preparation using Ovation Ultralow DR Multiplex System (0331, NuGEN Technologies, CA). Once prepared, library was sequenced using HiSeq2000 with single-end 40 nt read length.

## Sequential ChIP-qPCR

$3 \times 10^6$ of *Ehmt2*[F/−] *CreER*[+ve] day 2 EpiLCs were used per experiment. Cells were fixed and frozen as for lcChIP. Similarly, samples were lysed with 1ml lysis buffer. After pelleting the nuclei, they were resuspended in 1 ml dilution buffer and sonicated for lcChIP. Precleared lysates were immunoprecipitated overnight with anti-H3K9me2, anti-H3K27ac, or IgG controls immobilized with Dynabeads Antibody Coupling Kit. After performing six 10 min long washes (2× low salt, 2× high salt, 2× LiCl buffer), complexes were eluted in lysis buffer. Ten percent of each sample was saved for enrichment validation. Remaining samples were diluted ten-fold and used further in lcChIP-qPCR protocol.

## ChIP-seq analysis

ChIP-seq reads were extended to a total length of 250 nt for lcChIP, and 150 or 300 nt for H3K27me3 and H3K9me2 nChIP according to selected fragment sizes. Reads were aligned to the mouse reference genome (GRCm38/mm10) using bowtie with parameters '–m 1 –v 2' (*Langmead et al., 2009*). ChIP-seq intensities on all genomic regions (promoters, gene bodies, enhancer regions and 1kB tiles) were quantified as Log2(normaliseormalized ChIP/input) values. ChIP and input values were obtained as read counts per genomic region divided by total number of mapped reads and divided by the size of the region in kB. UCSC genome browser tracks represent Log2(normalised ChIP/input) on 200 nt-sliding windows with a 50 nt offset. To compare genome-wide distributions of H3K9me2 and H3K27me2 in different cell types, 1kB tiles were calculated for all chromosomes with a 500 bp offset, and each tile was intersected with annotated genomic regions obtained from the UCSC Table Browser.

Published ChIP-seq datasets were downloaded from GEO (GSE56138 ES 2i: p300, H3K27ac, H3K4me1, EpiLC: H3K27ac, H3K4me1 plus Activin conditions; GSE57409 EpiSC: H3K27ac, H3K4me1). To determine active enhancers in 2i/LIF ESCs, peaks were called for p300, H3K4me1,

and H3K27ac ChIP-seq datasets in ES 2i with MACS (liulab.dfci.harvard.edu/MACS). p300 peaks that intersected with an H3K27ac and H3K4me1 peak, but not with any annotated promoter region were classified as active enhancers.

To classify regions as marked by H3K9me2 and H3K27me3, we have used average Log2(normalised ChIP/input) from two biological replicates in all the analysis. Regions were separated into three clusters by k-means method and the most highly enriched was designated as marked. For differential enrichment analysis between EpiSCs and E6.25 epiblast, EdgeR was used (pval<0.05, Log2FC>2) (*Nikolayeva and Robinson, 2014*).

## Classification of epigenetic signatures by self-organizing maps

To classify epigenetic signatures of promoters and gene bodies, histone modification (Log2[ChIP/input]) and DNA methylation levels on each genomic region and expression levels of the corresponding genes were represented by n-dimensional feature vectors. Feature vectors were centre-scaled (mean=0, standard deviation=1), and self-organizing maps (SOMs) were trained using the R kohonen() package (*Wehrens and Buydens, 2007*). After training, each node on the hexagonal map represented a set of promoters or gene bodies with very similar epigenetic signatures and transcriptional states. The figures show the average epigenetic modifications and expression levels for the genomic regions summarized in each node. Neighbouring nodes on the map also contain genomic regions with similar signatures and expression levels as a result of an update function, which adjusts close neighbours with decreasing distance during training.

To classify enhancer signatures across different cell types, the same analysis was performed with feature vectors representing histone modifications and DNA methylation levels on enhancer regions (p300 peak summits ± 800 nt) in 2i/LIF ESCs, EpiLC, EpiSC and epiblast. For the representation of sets of enhancers with highly similar signatures as ranked heatmaps, the average values of the epigenetic modifications for the enhancers in each SOM node were determined and ranked by their average H3K9me2 values in EpiLCs or by their average H3K27ac values in EpiSC.

## DNA motif enrichment analysis

Eight hundred and forty-three position-specific weight matrices were determined by high-throughput SELEX sequencing for human and mouse DNA-binding domains (*Jolma et al., 2013*). Enhancer regions (p300 peak summits ± 800 nt) that were marked by H3K9me2 or H3K27me3 were scanned using fimo (http://meme-suite.org). To evaluate enrichment, 1000 sets of random regions of the same size were generated in gene regions (± 50 kB of flanking intergenic region), and the frequency of DNA-binding motifs was determined using a p-value threshold of 1e-5.

## WGBSeq

DNA was isolated from d2 EpiLC and EpiSC pellets using DNeasy blood and tissue kit (Qiagen, Germany). For every sample, 200 ng was sheared into ~250 bp using S220 Focused ultrasonicator (Covaris, MA). A total of 160 ng of sonicated and purified DNA was used for library preparation using Ovation Ultralow Methyl-Seq Library System (NuGEN Technologies, CA). This entailed a 3.5 h bisulfite conversion of adapter-ligated DNA using EZ DNA Methylation-Direct (Zymo Research, CA). The DNA was amplified using seven cycles. Once libraries were prepared, they were sequenced using HiSeq2000 with paired-end 100 nt read length. Reads were quality trimmed and aligned to the bisulfite-converted genome with Bismark with parameters –n –l 40 (*Krueger and Andrews, 2011*). Genome-wide DNA methylation levels and differentially methylated regions were determined by using MethPipe (*Song et al., 2013*). Published WGBS datasets were downloaded from the EBI Read Archive and from GEO (ERR192350) and processed by the same analysis pipeline.

## Acknowledgements

We are grateful to Alexander Tarakhovsky and Dónal O'Carroll for sharing G9a conditional knockout mice. We thank Jenny Nichols for critical input into the project and members of the Surani Lab for helpful discussions. We also thank Charles Bradshaw for bioinformatic support.

## Additional information

### Funding

| Funder | Grant reference number | Author |
|---|---|---|
| Wellcome Trust | WT096738 | Jan J Zylicz<br>Ufuk Günesdogan<br>Jamie A Hackett<br>Delphine Cougot<br>Caroline Lee<br>MA Surani |
| European Commission | | Ufuk Günesdogan |
| Wellcome Trust | RG44593 | Jan J Zylicz |

The funders had no role in study design, data collection and interpretation, or the decision to submit the work for publication.

### Author contributions

JJZ, Conception and design, Acquisition of data, Analysis and interpretation of data, Drafting or revising the article, Contributed unpublished essential data or reagents; SD, Analysis and interpretation of data, Drafting or revising the article; UG, DC, Drafting or revising the article, Contributed unpublished essential data or reagents; JAH, Conception and design, Drafting or revising the article; CL, Acquisition of data, Contributed unpublished essential data or reagents; MAS, Conception and design, Analysis and interpretation of data, Drafting or revising the article

### Author ORCIDs

Jan J Zylicz, http://orcid.org/0000-0001-9622-5658
M Azim Surani, http://orcid.org/0000-0002-8640-4318

### Ethics

Animal experimentation: All husbandry and experiments involving mice were authorized by a UK Home Office Project License 80/2637 and carried out in a Home Office-designated facility.

## Additional files

### Supplementary files

• Supplementary file 1. List of primers used in this study.

### Major datasets

The following datasets were generated:

| Author(s) | Year | Dataset title | Dataset URL | Database, license, and accessibility information |
|---|---|---|---|---|
| Zylicz J, Dietmann S, Günesdogan U, Hackett JA, Cougot D, Do DV, Lee C, Surani MA | 2015 | Chromatin dynamics and the role of G9a in gene regulation and enhancer silencing during early mouse development | http://www.ncbi.nlm.nih.gov/geo/query/acc.cgi?acc=GSE70355 | Publicly available at the NCBI Gene Expression Omnibus (Accession no: GSE70355). |

The following previously published datasets were used:

| Author(s) | Year | Dataset title | Dataset URL | Database, license, and accessibility information |
|---|---|---|---|---|
| Buecker C, Srinivasan R, Wu Z, Calo E | 2014 | Reorganization of enhancer patterns in transition from nave to primed pluripotency | http://www.ncbi.nlm.nih.gov/geo/query/acc.cgi?acc=GSE56138 | Publicly available at the NCBI Gene Expression Omnibus (Accession no: GSE56138). |

| Factor DC, Corradin O, Zentner GE, Saiakhova A | 2014 | Epigenomic comparison of distinct pluripotent stem cell states reveals a new class of enhancers with roles throughout mammalian development | http://www.ncbi.nlm.nih.gov/geo/query/acc.cgi?acc=GSE57409 | Publicly available at the NCBI Gene Expression Omnibus (Accession no: GSE57409). |
|---|---|---|---|---|
| Hayashi K, Kurimoto K, Ohta H, Saitou M | 2011 | Reconstitution of the mouse germ-cell specification pathway in culture by pluripotent stem cells | http://www.ncbi.nlm.nih.gov/geo/query/acc.cgi?acc=GSE30056 | Publicly available at the NCBI Gene Expression Omnibus (Accession no: GSE30056). |
| Boroviak T, Loos R, Bertone P, Smith A, Nichols J | 2014 | RNA sequencing of mouse embryonic day 3.5 blastocyst inner cell mass cells | http://www.ebi.ac.uk/ena/data/view/ERP005749 | Publicly available at the EBI European Nucleotide Archive (Accession no: ERP005749). |

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
