## [Decision Letter]

Thank you for submitting your work entitled "Chromatin dynamics and the role of G9a in gene regulation and enhancer silencing during early mouse development" for peer review at *eLife*. Your submission has been favorably evaluated by Fiona Watt (Senior editor) and three reviewers, one of whom is a member of our Board of Reviewing Editors.

The reviewers have discussed the reviews with one another and the Reviewing editor has drafted this decision to help you prepare a revised submission.

Essential revisions:

The three reviewers found your manuscript very interesting. However, there was consensus that currently the manuscript lacks coherence and would greatly benefit from removing parts of the data and strengthening other aspects.

Below, I summarize the main discussion points raised during the review process that we would like you to address upon revision.

In particular, we suggest that you could consider removing parts from the early embryo due to lack of functional relevance.

The reviewers were excited by the ChIP-seq data from epiblast, but felt that additional validations (see comments from Reviewer 2) and analyses (point 4 of Reviewer 3) would maximize the impact of the study.

There was also a suggestion to remove the p53 where indirect effects cannot be ruled out. However, it was also discussed that one value of the p53 ChIP result from G9a KO is demonstration that G9a loss enhances p53 binding at select enhancers, without appreciably affecting H3K27ac (Figure 6). The reviewers suspect that this may be an indirect consequence of either increased accessibility at enhancers associated with loss of H3K9me2 (which would be an interesting phenotype supporting role of G9a in enhancer repression) or simply of an increased cellular stress resulting in increase in p53 stability (perhaps less interesting). If combined with some accessibility data and eRNA expression analysis at matching enhancers (none of the enhancers at which eRNAs were analysed corresponds to the p53-bound enhancers from Figure 6), the p53 ChIP result could be used to strengthen the conclusions on the role of G9a in regulating enhancer function – which currently needs further work.

We also highly encourage you to further strengthen the computational analysis and specificities required. This should help in addressing a number of points raised by the three reviewers.

*Reviewer #1:*

In the current study, Zylicz and colleagues explore the role of G9a and EZH2 during early embryonic development encompassing the 2-cell stage and early blastocysts. The quantity and technical aspects of this study are very impressive. Each of the experiments provides important insights into the function of G9a and EZH2 in priming gene regulatory networks in epiblast cells. However, the experiments are disjointed and do not flow together well to make one coherent story. There is sufficient data here to produce two or more coherent stories exploring individual early findings reported by the authors. For instance, data from the 2-cell study and the transposons experiments can be removed to improve the flow of the paper as they detract from the main message of the study.

1) The H3K9me2 expression levels in Figure 1, compared to GLP and G9a expression levels in Figure 1–figure supplement 1 are standardized to different time points. In addition, different time points are presented (i.e. E3.5 versus E4.5; E5.5 versus E6.5). Is this the reason that expression of H3K9me2 and G9a/GLP does not directly correlate? The authors should standardize the expression to the same stage if they wish to draw comparisons.

2) It is not clear why the samples from the G9a M/Z and M/+ are pooled in the analysis presented in Figure 1 and Figure 1-figure supplement 2E? In other panels the data is separated. Transcription of some genes from the zygotic genome has been reported as early as the 2 cell stage and hence, these data should be separated. This may also reduce the large variation seen in Figure 1 and Figure 1–figure supplement 2E.

3) During the RNA seq. analysis, why was a fold change cut-off of log2(1.6) used? Furthermore, was a false discovery rate, which is more commonly used, applied to the data? How can the authors rule out that all direct targets of G9a are changed by at least log2(1.6)?

4) In the *G9a^-/-^* versus control RNA seq., G9a mRNA does not appear to be reduced ([Supplementary-material SD1-data]). Is there a reason for this discrepancy?

5) What was the cut-off for significance in the *Ezh2^-/-^* RNA Seq. dataset? It is not clear in the paper. Is it the same as the *G9a^-/-^* RNA Seq? The authors need to make this information clear.

6) Figure 3 shows only a small number of transcriptional changes despite H3K27me3 and H3K9me2 being wide spread. Does the correlation change if the arbitrary log2 cut-off of 1.6 is changed? Deletion of chromatin modifying proteins does not always result in very large changes in the expression of target genes.

7) The authors need to further address the discrepancy in OCT4 level between their study and Yamamizu et al., 2012. It seems odd that there is such a large incongruity between the two studies. Only one line is provided in the Discussion on this topic and it does not sufficiently propose any reasons for such discrepancy. Given that lack of changes in pluripotency is central to the conclusions of this paper, the authors should undertake staining for OCT4 in *G9a^-/-^*embryos and controls.

8) The authors show in Figure 2 that p21 is upregulated in a subset of *G9a^-/-^*samples, which is consistent with increased H3K9me2 at the p21 locus during ESC differentiation. The authors claim that these observations support their view that the delayed development of *G9a^-/-^* embryos is related to increased apoptosis and/or reduced proliferation, and not retainment of pluripotency. The authors have provided little evidence for this. The authors should validate their proposal in vivo by undertaking BrdU incorporation studies and phospho-H3 staining for proliferation, as well as TUNEL or active caspase-3 staining to analyse cell death.

9) In addition, can the authors exclude that the expressions of important developmental genes are normal at E6.5? For instance, a very high level of Bmp4 is observed in *G9a^-/-^* E6.25 epiblasts (Figure 2), which could be one explanation for developmental defects. In situ hybridizations for developmentally important genes including Bmp4 need be carried out.

10) In the E6.25 RNA Seq. data ([Supplementary-material SD1-data]), p21 mRNA is not reduced in *G9a^-/-^* epiblasts. Can the authors explain its absence?

11) Since the authors state that a significant proportion of genes affected by G9a deletion are involved in germ line specification, the authors need to show whether germ cells are affected. Are PRDM14 and Blimp1 expression patterns normal in *G9a^-/-^* embryos? If pluripotency is indeed unaffected, while germ line specification is presumably impaired, can the authors show normal alkaline phosphatase staining in *G9a^-/-^* embryos and not in germ cells?

12) The authors claim that 2i ESCs as primitive ectoderm (PrE). In contrast, the works from Austin Smith's lab and others (Van Oosten et al., 2012 for example) have shown that the ESC state in 2i resembles the naïve state of the epiblast. Why do the authors consider the 2i state to be similar to the PrE? Rather, ESCs in standard ESC medium (with serum) are considered to be more like E4.5 PrE.

*Reviewer #2:*

Please note that I cannot judge the stringency of the bioinformatics pipelines used in depth, and therefore my comments are focused mostly on the biology of the questions addressed and the conclusions drawn, rather than in the stringency of the computational side. I do find, in general, that the computational data is not always presented in a manner for a broad readership, and some points below suggest improvements towards this direction.

Although a few of the observations documented are not particularly new, the manuscript by Zylics and colleagues builds on former data on Ezh2 and G9a in regulating development around implantation. The strongest point in, my view, is the analysis of epiblast cells in vivo at the transcriptional level and at the chromatin level (K9me2 and K27me3) genome wide. The experiments are carefully executed and in most instances the conclusions drawn are supported by the data presented.

I have, however, mixed feelings about the presentation and the flow of the manuscript. It does not come out as a single, solid message, and is diluted with many pieces of data, some of which I find irrelevant for the main conclusions and I would therefore suggest to remove these parts, which encompass all the pre-implantation data (which is not strong enough), as well as the p53 data at the end, which in the end are not very strong. This would allow the authors to concentrate in a less dense manner, on the main messages of the paper based on the transitions between naïve/primed and epics states.

I have some comments on the ChIP protocol validation that require in my view, additional experimental work, and some comments to draw to the phrasing and statistics throughout the manuscript, as well as clarification or additional computational analysis.

The 'pregastrulation conclusion’ (paragraph four of the Discussion) is reasonable, but the preimplantation work is confusing, and poorly documented in comparison with the peri-implantation and in vitro data (e.g. statement of 2C specific gene regulation by G9a is based on only 3 genes) and does not add much to the manuscript and instead deviates the focus of attention from the pre-gastrulation cell fate transitions.

All figures lack description of quantification of fluorescence intensities, as well as the N numbers for IF and quantifications. The only description is 'fluorescence intensity was normalised to DAPI', but the authors state that they used 'quantitative immunofluroescence' to analyse levels. Please add a better description and/or controls for this.

In Figure 3: Why is ChIP enrichment mostly negative – it suggests rather a depletion of mark? Is there any enrichment at all e.g. for K9me2 in Figure 3? The same applies for Figure 3—figure supplement 2 C, in which H3k9me2 'enrichment' is again below 0.

From Figure 3, the authors conclude that K9me2 and K27me3 are on different regions. However, the data presented do not fully support these conclusions, both from analysis in 3B and from the self-organising maps: while there are some regions that are not overlapping, there are clearly other regions that do show overlap. The shape of the line in the fitting in 3B also suggests this interpretation. Since this analysis was not presented with statistics, nor with a comparison of e.g. active promoters or full genome data for 3B, I believe that the authors are not in the position to conclude what they conclude, so this part needs reinterpretation.

Regarding my validation point above, the only validation presented for the lcChip-seq is a comparison of Pearson figures (Figure 3—figure supplement 1) based on clustering of independent replicates – what if replicates are all poor showing similar background enrichment? The protocol would need validation to assess a comparison in magnitude of enrichment to standard protocol. I assume the authors have some material left to run a couple of PCR reactions, can they estimate by ChIP-PCR the differences in order of magnitude with the lcChIP in EpiSC versus the 'normal' EpiSC ChIP protocol?

I do not see much significance/added value on the p53 data (in the subsection “G9a Mediates Efficient Enhancer Inactivation”), and it only adds more 'density' in the manuscript, which is quite dense. Specially, how relevant (biologically and statistically) is the "15%" of H3K9me2 and H3K27ac enhancers being bound by p53?

The analysis of 'proximity' of eRNA is not presented adequately, nor it is known what 'proximity' is? In kb? This cannot be inferred from the graphs presented. Can the authors do an enrichment analysis of eRNA expression (on the y axis) relative to specific distances of genes (on the x axis, in kb). The conclusion on page 15 on H3K9me2 domains 'extend to active enhancers targeted for silencing, which accounts for coenrichment with H3K27ac' implies a causal relationship for which there is no data.

*Reviewer #3:*

In this study Zylicz et al. set out to explore the deposition and function of repressive histone marks such as H3K27me3 and H3K9me2 during early embryonic development. The authors generated embryos deficient for G9a or Ezh2 methyltransferases and examined their gene expression profile using single epiblast RNA-seq at day E6.25, prior to the overt phenotypic manifestations. To determine which genomic loci are marked by repressive histone methylation marks in vivo, the authors performed low cell number ChIP-seq assays for H3K27me3 and H3K9me2 in E6.25 epiblasts. Strikingly, for either mark, an exceedingly small subset of marked genes is perturbed upon loss of respective enzyme, suggesting either existence of multiple redundant silencing mechanisms or these methylation events being a secondary consequence of repression. Furthermore, the set of genes that are aberrantly upregulated are non-overlapping between the two different knock outs and show therefore that in the epiblast H3K27me3 and H3K9me2 are involved in repression of different set of genes. The authors note that many intergenic enhancer regions are marked by H3K9me2 in the epiblast, and they follow up upon these observations using the in vitro cellular models. They conclude that G9a mediates efficient inactivation of a subset of enhancers, particularly those bound by p53.

This study contains many interesting observations, tackles an important biological question and is generally well-executed. However, it also suffers from weaknesses that dampen my overall enthusiasm, but which could be fairly easily addressed.

1) The authors claim that a subset of enhancers gains H3K9me2 in epiblast, and that this is associated with their inactivation during transition to primed pluripotency. Several issues need to be clarified here:

In Figure 4, heatmaps of H3K9me2 from ESC (to show that these loci indeed gain H3K9me2 during differentiation) and 6.5 epiblasts (to show that similar findings can be observed in vivo) should be included. Furthermore, how would these heatmaps look if authors chose as their analysed regions enhancers that are active in EpiLC/EpiSC, but not in ESC? What would be the corresponding profiles (if any) of H3k9me2 status at these regions?

2) A substantial weakness of the study is that despite the authors having all the necessary reagents in hand, the data pertaining to the functional impact of G9a on enhancer activity is limited to just a few examples, and not coherent ones either (e.g. enhancers shown in Figure 6 don't overlap those shown in Figure 6). Given that the role of G9a in enhancer inactivation is one of the central novel claims of the paper, at the very least the eRNA expression analysis from the *G9a^F/–^*CreER EpiLCs treated with EtOH or TAM should be done systematically, across all H3K9me2 marked enhancers, and in comparison to enhancers unmarked by H3K9me2, but containing similar enrichment levels of H3K27ac. The relationship between enhancers with G9a-dependent eRNA expression and transcriptional changes at neighboring genes should be determined. To what extent transcriptional changes observed in G9a KO cells can be explained by H3K9me2/eRNA changes at nearby enhancers?

3) Increased p53 binding upon loss of G9a may be an indirect consequence of the defect in enhancer repression, perhaps resulting in the increased enhancer accessibility- did the authors consider looking at chromatin accessibility measurements such as ATAC, DNase or FAIRE? Such data could in fact strengthen the conclusions on the functional impact of G9a on enhancer activity, regardless of the specific impact on p53.

4) The authors generate valuable H3K9me2 and H3K27me3 ChIP-seq datasets from epiblast cells of E6.25 embryos, but I feel they missed the opportunity to provide a deeper comparison of their results with the H3K9me2 and H3K27me3 patterns reported from the in vitro cellular models such as EpiLC or EpiSC. What are the major similarities and differences in patterns? A systematic (and to whatever extent possible, quantitative) comparison of H3K9me2 patterns across the genome in different cellular states would not only be broadly interesting for the community, but also relevant for the authors' own conclusions.

---

## [Author Response]

*The three reviewers found your manuscript very interesting. However, there was consensus that currently the manuscript lacks coherence and would greatly benefit from removing parts of the data and strengthening other aspects.*

To address this point, we have removed all observations on the effects of G9a mutation on preimplantation embryos, and focused on the transition from naïve to primed pluripotency. We also performed additional experiments on the link between p53 and G9a. Since we found an increase of activated phopho-p53 globally in cells lacking G9a (see Figure 10), we cannot determine the effects of H3K9me2 on recruitment of p53 to enhancers. Accordingly and in response to Reviewer #2, we have removed the p53-related analysis from the paper.

Author response image 1.Western Blot of d2 *G9a^F/-^ CreER^+ve^* EpiLCs treated with ethanol (EtOH) or tamoxifen (TAM).Membranes were probed with anti-G9a, anti-phopho-p53 and anti-aTubuline antibodies.**DOI:**
http://dx.doi.org/10.7554/eLife.09571.032

As suggested by the reviewers, we have included additional bioinformatics analysis for detailed comparison between in vitro EpiSCs and in vivo epiblast (Figure 4 and Figure 4—figure supplement 1). We have also added whole-genome bisulfite sequencing dataset performed on EpiSC, which reveals substantial DNA hypermethylation compared to the epiblast. To increase the depth of analysis, we have incorporated published RNAseq analysis of E3.5 ICM (Boroviak et al., 2014), which has enabled analysis of ChIPseq data in the context of developmental progression (Figure 3, Figure 3—figure supplement 3).

In response to the reviewers’ suggestion, we have expanded our analysis of in vivo IF, which convincingly shows accumulation of H3K9me2, GLP and G9a upon implantation (Figure 1). We also reveal an increase in the number of apoptotic and non-proliferative cells as judged by cleaved caspase 3 and Ki67 staining (Figure–figure supplement 2C, D).

*In particular, we suggest that you could consider removing parts from the early embryo due to lack of functional relevance.*

Indeed we agree and removed this part.

*The reviewers were excited by the ChIP-seq data from epiblast, but felt that additional validations (see comments from Reviewer 2) and analyses (point 5 of Reviewer 3) would maximize the impact of the study.*

This ChIP-seq dataset is indeed novel, and now we have included additional validation by ChIP-qPCR (Figure 3—figure supplement 1), which shows that low cell number ChIP maintains robust enrichment over control regions and thus is suitable for genome-wide analysis. We also include additional bioinformatic analysis:

Detailed comparison between EpiSCs and in vivo epiblast including WGBSeq dataset (Figure 4, Figure 4—figure supplement 1).

Analysis of ChIPseq in the context of exit from naïve pluripotency by incorporating published E3.5 RNAseq datasets (Figure 3, Figure 3—figure supplement 3).

Inclusion of an epiblast sample into the enhancer analysis (Figure 6) as well as extension of such analysis to EpiLC/EpiSC active enhancers (Figure 6—figure supplement 1).

The additional analysis shows that EpiSCs accumulate aberrant DNA methylation and loose H3K27me3 from germline genes, which might limit their competence for germline fate. By including published E3.5 ICM dataset, we are able to identify differentially expressed genes during the exit from naïve pluripotency. In this context we reveal that H3K27me3 and H3K9me2 target specific pathways independently. Finally, accumulation of H3K9me2 in EpiLCs enhancers is also seen in the in vivo epiblast.

*There was also a suggestion to remove the p53 where indirect effects cannot be ruled out. However, it was also discussed that one value of the p53 ChIP result from G9a KO is demonstration that G9a loss enhances p53 binding at select enhancers, without appreciably affecting H3K27ac (Figure 6). The reviewers suspect that this may be an indirect consequence of either increased accessibility at enhancers associated with loss of H3K9me2 (which would be an interesting phenotype supporting role of G9a in enhancer repression) or simply of an increased cellular stress resulting in increase in p53 stability (perhaps less interesting). If combined with some accessibility data and eRNA expression analysis at matching enhancers (none of the enhancers at which eRNAs were analysed corresponds to the p53-bound enhancers from Figure 6), the p53 ChIP result could be used to strengthen the conclusions on the role of G9a in regulating enhancer function – which currently needs further work.*

We show that loss of G9a in EpiLCs leads to increased activation of p53 globally as judged by western blot for phopho-p53 (Figure 10), resulting in increased p53 stability upon loss of G9a. For this reason, we have removed the data on the recruitment of p53 to G9a bound enhancers as it is most likely an indirect effect. The enhancer experimental part now shows a consistent panel of 13 enhancers that were analysed with respect to eRNA expression dynamics. We show H3K9me2 accumulation, H3K27ac enrichment, sequential ChIPqPCR and eRNA misexpression in G9a mutant cells in vitro and in vivo. The strength of this comprehensive analysis is that it looks directly at the epigenetic and transcriptional state of enhancers rather than relying on a possible erroneous assumption that enhancers regulate the nearest genes. Nevertheless, we extracted the expression of nearby genes from E6.25 epiblasts and can confirm that 3/8 genes in proximity of H3K9me2 marked enhancers are indeed upregulated (Figure 11). However, we prefer not to include this analysis in the publication, as we are not able to reliably link the enhancers to specific genes.

Author response image 2.Bar plot showing changes in expression levels of genes nearest to analysed enhancers in *G9a^cntr^*and *G9a^-/-^* E6.25 epiblast.Data was extracted from RNAseq experiment. Shown are average Log2(Fold change) (+/- SEM). * pvalue<0.05 using Wilcoxon rank-sum test**DOI:**
http://dx.doi.org/10.7554/eLife.09571.033

*We also highly encourage you to further strengthen the computational analysis and specificities required. This should help in addressing a number of points raised by the three reviewers.*Reviewer #1:

*In the current study, Zylicz and colleagues explore the role of G9a and EZH2 during early embryonic development encompassing the 2-cell stage and early blastocysts. The quantity and technical aspects of this study are very impressive. Each of the experiments provides important insights into the function of G9a and EZH2 in priming gene regulatory networks in epiblast cells. However, the experiments are disjointed and do not flow together well to make one coherent story. There is sufficient data here to produce two or more coherent stories exploring individual early findings reported by the authors. For instance, data from the 2-cell study and the transposons experiments can be removed to improve the flow of the paper as they detract from the main message of the study.*

We thank the reviewer for the supportive comments. We have now removed both preimplantation experiments as well as p53 ChIP. Instead we concentrate on in-depth analysis of H3K9me2 and H3K27me3 during the transition from naïve to primed pluripotent cells.

*1) The H3K9me2 expression levels in Figure 1, compared to GLP and G9a expression levels in Figure 1–figure supplement 1 are standardized to different time points. In addition, different time points are presented (i.e. E3.5 versus E4.5; E5.5 versus E6.5). Is this the reason that expression of H3K9me2 and G9a/GLP does not directly correlate? The authors should standardize the expression to the same stage if they wish to draw comparisons.*

We have now removed most of the data on preimplantation IF staining and instead concentrated on two crucial stages (E3.5 and E5.5), which is now part of updated Figure 1. Original inconsistencies occurred because different antibodies showed excessive background staining in the visceral endoderm. To circumvent this problem, we had to remove visceral endoderm at E5.5 for staining with anit-G9a antibodies.

*2) It is not clear why the samples from the G9a M/Z and M/+ are pooled in the analysis presented in Figure 1 and Figure 1–figure supplement 2E? In other panels the data is separated. Transcription of some genes from the zygotic genome has been reported as early as the 2 cell stage and hence, these data should be separated. This may also reduce the large variation seen in Figure 1 and Figure 1–figure supplement 2E.*

We have decided to remove all observations on preimplantation G9a mutants from this manuscript.

*3) During the RNA seq. analysis, why was a fold change cut-off of log2(1.6) used? Furthermore, was a false discovery rate, which is more commonly used, applied to the data? How can the authors rule out that all direct targets of G9a are changed by at least log2(1.6)?*

In this experiment, we used a low input protocol for single epiblast RNAseq. There is some biological variability between the control embryos as well as between KO embryos. We take that into account by using pairwise comparison between individual epiblasts to make sure that a minimum threshold is met in all control vs KO comparisons. Indeed, setting thresholds is to some extent arbitrary but is commonly used in transcriptional analysis. In Figure 3 of the Response document, we show estimated rate of false positives in relation to the threshold used. This rate has been estimated based on the biological and technical variability observed between control samples. At a Log2(1.4) the ratio of identified genes versus false positives reaches 0.01. The same threshold has been set for both G9a and Ezh2 RNAseq experiments.

*4) In the* G9a^-/-^
*versus control RNA seq., G9a mRNA does not appear to be reduced ([Supplementary-material SD1-data]). Is there a reason for this discrepancy?*

G9a is also called *Ehmt2*, which is cited in the source data file and is indeed significantly down-regulated. To avoid confusion, we have now updated the name of the gene to *Ehmt2* in the whole manuscript.

*5) What was the cut-off for significance in the* Ezh2^-/-^
*RNA Seq. dataset? It is not clear in the paper. Is it the same as the* G9a^-/-^
*RNA Seq? The authors need to make this information clear.*

We now make it clear in the text that the parameters used were identical to those for G9a RNAseq (paragraph three, subsection “Epigenetic Programming Regulates Growth and Development of the Embryo”).

*6) Figure 3 shows only a small number of transcriptional changes despite H3K27me3 and H3K9me2 being wide spread. Does the correlation change if the arbitrary log2 cut-off of 1.6 is changed? Deletion of chromatin modifying proteins does not always result in very large changes in the expression of target genes.*

Indeed it is true that small transcriptional changes are also possible. Such effects however are confounded by the variability between control epiblasts and technical variability introduced by using a low input RNAseq protocol. Hence we are unable to rigorously estimate and identify such transcripts. In Figure 12 we show the number of genes identified in relation to the cut-off. At the lower end of the threshold the false discovery rate however is too large.

Author response image 3.Plot showing background (expected) and observed number of transcripts reaching specific fold change (FC) deregulation thresholds.Expected values were estimated based on variability of gene expression between *G9ac^ntr^* or *Ezh2^cntr^*RNAseq samples. Observed dataset are genes reaching FC threshold in *G9a^-/-^* or *Ezh2^-/-^* RNAseq samples. The ration of observed vs expected reaches 0.01 value only at log2(FC)>1.4.**DOI:**
http://dx.doi.org/10.7554/eLife.09571.034

*7) The authors need to further address the discrepancy in OCT4 level between their study and Yamamizu et al., 2012. It seems odd that there is such a large incongruity between the two studies. Only one line is provided in the Discussion on this topic and it does not sufficiently propose any reasons for such discrepancy. Given that lack of changes in pluripotency is central to the conclusions of this paper, the authors should undertake staining for OCT4 in* G9a^-/-^
*embryos and controls.*

We agree with the reviewer that IF would be relevant. We now show that GFP signal from Oct4-GFP does not show any significant differences between *G9ac^ntr^* and KO epiblasts (Figure 2—figure supplement 2). More importantly we focused on NANOG, which at E6.5 is already substantially down-regulated in the anterior epiblast (Osorno et al., 2012). Our whole-mount staining revealed comparable levels of NANOG in G9aKO and *G9ac^ntr^* epiblast (Figure 2—figure supplement 2 A).

*8) The authors show in Figure 2 that p21 is upregulated in a subset of* G9a^-/-^*samples, which is consistent with increased H3K9me2 at the p21 locus during ESC differentiation. The authors claim that these observations support their view that the delayed development of* G9a^-/-^
*embryos is related to increased apoptosis and/or reduced proliferation, and not retainment of pluripotency. The authors have provided little evidence for this. The authors should validate their proposal in vivo by undertaking BrdU incorporation studies and phospho-H3 staining for proliferation, as well as TUNEL or active caspase-3 staining to analyse cell death.*

In Figure 2—figure supplement 1 (C, D) we now show whole-mount IF staining against cleaved caspase 3 and Ki67. By carefully quantifying IF signal from 5 *G9a^+/+^* and 5 *G9a^-/-^* embryos we confirm increased rate of apoptosis in E6.5 mutant embryos. Indeed we also observed increased numbers of non-proliferative Ki67+ve cells. This indicates that loss of G9a results in increased apoptosis and reduced proliferation.

*9) In addition, can the authors exclude that the expressions of important developmental genes are normal at E6.5? For instance, a very high level of Bmp4 is observed in* G9a^-/-^
*E6.25 epiblasts (Figure 2), which could be one explanation for developmental defects. In situ hybridizations for developmentally important genes including Bmp4 need be carried out.*

In this study we show that loss of G9a leads to developmental delay, reduced proliferation and increased apoptosis. We report that there is no delay in the exit from pluripotency but there is upregulation of multiple developmentally relevant genes as well as some repeat elements. We have added observations on Nanog and Oct4-GFP, which remain unchanged in mutant embryos. Unfortunately while it is not possible to validate all RNAseq findings, we have however undertaken the task of doing so for some germline and proliferation genes (Figure 2). A potential increase in Bmp4 could contribute to the phenotype, but it is perhaps equally or more likely that epigenetic changes associated with the enhancers and promoters have important consequences for changes in gene expression. In our opinion, in situ hybridizations would not provide further insights on the relative contributions of signalling and cell autonomous changes to the phenotype. This aspect however is beyond the scope of this study.

*10) In the E6.25 RNA Seq. data ([Supplementary-material SD1-data]), p21 mRNA is not reduced in* G9a^-/-^
*epiblasts. Can the authors explain its absence?*

p21 is also called *Cdkn1a* it is under this name in the source data file and is indeed significantly up-regulated. We have updated the name for *Cdkn1a* in the whole manuscript.

*11) Since the authors state that a significant proportion of genes affected by G9a deletion are involved in germ line specification, the authors need to show whether germ cells are affected. Are PRDM14 and Blimp1 expression patterns normal in* G9a^-/-^
*embryos? If pluripotency is indeed unaffected, while germ line specification is presumably impaired, can the authors show normal alkaline phosphatase staining in* G9a^-/-^
*embryos and not in germ cells?*

Here we show that loss of G9a results in upregulation of multiple late germline cell fate regulators. However RNAseq experiments showed no difference with respect to *Prdm14* or *Blimp1*. Based on this observation, we posit that germline specification will remain unaffected. To address this issue, we collected severely delayed *G9a^-/-^ ΔPE-Oct4-GFP+ve E8.5* embryos. Whole-mount staining clearly showed a small population of *GFP^+ve^* and *AP2g^+ve^* putative PGCs at the base of allantois (Figure 2—figure supplement 2 A). The numbers of PGCs were however reduced in the mutants, probably because of their severe developmental delay. Thus, loss of G9a is compatible with germline specification despite premature expression of late germline markers.

*12) The authors claim that 2i ESCs as primitive ectoderm (PrE). In contrast, the works from Austin Smith's lab and others (Van Oosten et al., 2012 for example) have shown that the ESC state in 2i resembles the naïve state of the epiblast. Why do the authors consider the 2i state to be similar to the PrE? Rather, ESCs in standard ESC medium (with serum) are considered to be more like E4.5 PrE.*

A recent study from Austin Smith’s lab shows that 2i/LIF ESC might be more similar to primitive ectoderm of E4.5 embryos (Boroviak, 2014). However there is no direct genome-wide comparison between 2i/LIF ESCs, and different in vivo stages. Since there is also a lack of rigorous terminology in the field, we have decided to consistently use the most frequently used term: ICM (inner cell mass).

Reviewer #2:

*[…] I have, however, mixed feelings about the presentation and the flow of the manuscript, which does not come out as a single, solid message, and is diluted with many pieces of data, some of which I find irrelevant for the main conclusions and I would therefore suggest to remove these parts, which encompass all the pre-implantation data (which is not strong enough), as well as the p53 data at the end, which in the end are not very strong. This would allow the authors to concentrate in a less dense manner, on the main messages of the paper based on the transitions between naïve/primed and epics states.*

We thank the reviewer for a positive feedback about the quality of work presented and its interest for a broad readership. We have addressed a number of issues to make the manuscript more coherent. In particular, we have removed the work on preimplantation G9a mutant embryos as well as the p53-related work, and concentrated our attention on the transition between naïve and primed pluripotency in vivo. In this respect we have included an analysis of E3.5 ICM RNAseq. This allowed us to identify specific pathways repressed upon implantation and the epigenetic mechanisms involved in this process. We now show that regulators of metabolism and transcription are preferentially targeted for H3K27me3 accumulation. On the other hand H3K9me2 is linked to regulators of meiosis. We also provide comparison between in vivo epiblast and EpiSCs on the epigenetic level. This revealed substantial accumulation of DNA methylation in EpiSCs especially at germline related promoters, concomitantly with the loss of H3K27me3. This observation is consistent with the diminished competence of EpiSCs for PGC fate.

*The 'pregastrulation conclusion’ (paragraph four of the Discussion) is reasonable, but the preimplantation work is confusing, and poorly documented in comparison with the peri-implantation and in vitro data (e.g. statement of 2C specific gene regulation by G9a is based on only 3 genes) and does not add much to the manuscript and instead deviates the focus of attention from the pre-gastrulation cell fate transitions.*

We have decided to remove all work on preimplantation G9a mutants from this manuscript.

*All figures lack description of quantification of fluorescence intensities, as well as the N numbers for IF and quantifications. The only description is 'fluorescence intensity was normalised to DAPI', but the authors state that they used 'quantitative immunofluroescence' to analyse levels. Please add a better description and/or controls for this.*

This is now addressed in the revised manuscript. Every IF experiment was performed in at least 3 biological replicates, which is clearly stated in figure legends, including the numbers of embryos analysed, and how the signal was measured. For Figure 1, signal from individual ICM/Epiblast nuclei was measured and normalised to the intensity of DAPI signal. Such measurements were done on individual z-stacks in at least 3 embryos and 20 cells per developmental stage.

*In Figure 3: Why is ChIP enrichment mostly negative – it suggests rather a depletion of mark? Is there any enrichment at all e.g. for K9me2 in Figure 3? The same applies for Figure 3—figure supplement 2 C, in which H3k9me2 'enrichment' is again below 0.*

(This data is now presented in Figure 3 as well as Figure 3—figure supplement 2.)

We agree with the reviewer that this scale might be misleading. However, H3K9me2 does not form peaks but rather occupies large domains of enrichment where gene bodies and promoters are generally depleted of the mark compared to intragenic regions. Thus all elements reaching log2(ChIP/input)>0 are actually significantly enriched when compared to all genic regions as they reach levels typical for silent intergenic regions.

*From Figure 3, the authors conclude that K9me2 and K27me3 are on different regions. However, the data presented do not fully support these conclusions, both from analysis in 3B and from the self-organising maps: while there are some regions that are not overlapping, there are clearly other regions that do show overlap. The shape of the line in the fitting in 3B also suggests this interpretation. Since this analysis was not presented with statistics, nor with a comparison of e.g. active promoters or full genome data for 3B, I believe that the authors are not in the position to conclude what they conclude, so this part needs reinterpretation.*

(This data is now presented in Figure 3).

We have strengthened this result further by providing additional statistical analysis, and described this in the text (in the subsection “H3K9me2 and H3K27me3 are Associated with Distinct Repressive Chromatin States in vivo”). Classification of all promoters reveals that only ~0.3% show dual H3K9me2 and H3K27me3 marking. This result is consistent with a high level of anti-correlation that is statistically significant with p-value of 0.0024 (Chi^2^ test). We have also performed an analysis of H3K9me2 vs H3K27me3 at all promoters (Figure 13). However we do not include it in the manuscript as addition of active genes provides a population of H3K9me2-negative and H3K27me3-negative regions, it is thus impossible to plot a correlation curve.

Author response image 4.Density contour plots showing correlation between H3K27me3 and H3K9me2 enrichments at all promoters in E6.25 epiblast.**DOI:**
http://dx.doi.org/10.7554/eLife.09571.035

*Regarding my validation point above, the only validation presented for the lcChip-seq is a comparison of Pearson figures (Figure 3—figure supplement 1) based on clustering of independent replicates – what if replicates are all poor showing similar background enrichment? The protocol would need validation to assess a comparison in magnitude of enrichment to standard protocol. I assume the authors have some material left to run a couple of PCR reactions, can they estimate by ChIP-PCR the differences in order of magnitude with the lcChIP in EpiSC versus the 'normal' EpiSC ChIP protocol?*

In line with the reviewer’s comment we have now included a direct comparison of enrichments at control regions achieved using standard and low cell number ChIP protocol (Figure 3—figure supplement 1). This analysis shows only slightly reduced fold enrichment for H3K27me3 ChIPqPCR and no such effect for H3K9me2. This confirms that the low input method is suitable for genome-wide analysis.

*I do not see much significance/added value on the p53 data (in the subsection “G9a Mediates Efficient Enhancer Inactivation”), and it only adds more 'density' in the manuscript, which is quite dense. Specially, how relevant (biologically and statistically) is the "15%" of H3K9me2 and H3K27ac enhancers being bound by p53?*

We can confirm that a significant proportion of H3K9me2 enriched enhancers are also enriched for p53. However we have also found that loss of G9a leads to global activation of the p53 pathway. Thus increased recruitment of p53 to enhancers in *G9a^-/-^* cells is most likely an indirect effect. We decided to remove this part, and instead focus on the transition from naïve to primed pluripotency in vivo.

The analysis of 'proximity' of eRNA is not presented adequately, nor it is known what 'proximity' is? In kb? This cannot be inferred from the graphs presented. Can the authors do an enrichment analysis of eRNA expression (on the y axis) relative to specific distances of genes (on the x axis, in kb).

We have now updated this information, which is now in Figure 8. We include the distance of the enhancer to nearest gene’s transcriptional start site. However we have seen no correlation between the expression of eRNA and distance to the nearest gene and thus decided not to include it in the manuscript.

*The conclusion on page 15 on H3K9me2 domains 'extend to active enhancers targeted for silencing, which accounts for coenrichment with H3K27ac' implies a causal relationship for which there is no data.*

Indeed H3K9me2 accumulation does not seem to be actively targeted to specific loci, but rather extends to regions silenced or undergoing repression. However, in our view this phrase does not imply causal relationship.

Reviewer #3:

This study contains many interesting observations, tackles an important biological question and is generally well-executed. However, it also suffers from weaknesses that dampen my overall enthusiasm, but which could be fairly easily addressed.

*1) The authors claim that a subset of enhancers gains H3K9me2 in epiblast, and that this is associated with their inactivation during transition to primed pluripotency. Several issues need to be clarified here:*

*In Figure 4, heatmaps of H3K9me2 from ESC (to show that these loci indeed gain H3K9me2 during differentiation) and 6.5 epiblasts (to show that similar findings can be observed in vivo) should be included.*

(This data is now presented in Figure 6).

To address this question, we performed a more in-depth analysis of enhancer dynamics between different cell types. We have classified all active ESC enhancers based on the H3K27ac/H3K9me2 status in ESC/EpiLC/EpiSC and clustered similar elements together (Figure 6). This heatmap now includes ESC samples and shows striking gain of H3K9me2 in primed cells. Moreover H3K9me2-marked enhancers show loss of H3K27ac in EpiSC. Finally we have included E6.25 epiblast sample to show that the same enhancers gain H3K9me2 in vivo.

*Furthermore, how would these heatmaps look if authors chose as their analysed regions enhancers that are active in EpiLC/EpiSC, but not in ESC? What would be the corresponding profiles (if any) of H3k9me2 status at these regions?*

This point is directly addressed in Figure 6—figure supplement 1 C. Again we have clustered enhancers but as suggested by the reviewer we have included regions active in EpiLC and EpiSC. Next we sorted the clusters based on the H3K27ac level in EpiSC. This showed that highly active enhancers in EpiSC are devoid of H3K9me2. This result confirms that H3K9me2 is excluded from active enhancer in self-renewing cells.

*2) A substantial weakness of the study is that despite the authors having all the necessary reagents in hand, the data pertaining to the functional impact of G9a on enhancer activity is limited to just a few examples, and not coherent ones either (e.g. enhancers shown in Figure 6 don't overlap those shown in Figure 6). Given that the role of G9a in enhancer inactivation is one of the central novel claims of the paper, at the very least the eRNA expression analysis from the* G9a^F/*–*^*CreER EpiLCs treated with EtOH or TAM should be done systematically, across all H3K9me2 marked enhancers, and in comparison to enhancers unmarked by H3K9me2, but containing similar enrichment levels of H3K27ac. The relationship between enhancers with G9a-dependent eRNA expression and transcriptional changes at neighboring genes should be determined. To what extent transcriptional changes observed in G9a KO cells can be explained by H3K9me2/eRNA changes at nearby enhancers?*

We have now removed all p53 bound enhancers from the analysis as their increased activation might be linked to indirect p53 stabilisation. Therefore, measuring eRNA expression at these loci would not be informative for this. The enhancer analysis now shows a consistent panel of 13 regions that were analysed with respect to eRNA expression dynamics, H3K9me2 accumulation, H3K27ac enrichment, sequential ChIPqPCR and eRNA misexpression in G9a mutant cells in vitro and in vivo. The strength of this comprehensive analysis is that it looks directly at the epigenetic and transcriptional state of enhancers rather than relying on a potentially erroneous assumption that enhancers regulate the nearest genes. Nevertheless, we have extracted the expression of nearby genes from RNAseq on E6.25 epiblasts and can confirm that 3/8 genes in proximity of H3K9me2 marked enhancers are indeed upregulated in *G9a^-/-^* embryos (Figure 11). However, we have not included this analysis in the paper as we are not able to reliably link enhancers with specific genes. Finally, assessing the expression of nearby genes is further confounded by the fact that they themselves might be targets of H3K9me2 at promoters or gene bodies.

*3) Increased p53 binding upon loss of G9a may be an indirect consequence of the defect in enhancer repression, perhaps resulting in the increased enhancer accessibility- did the authors consider looking at chromatin accessibility measurements such as ATAC, DNase or FAIRE? Such data could in fact strengthen the conclusions on the functional impact of G9a on enhancer activity, regardless of the specific impact on p53.*

We have now found that increased recruitment of p53 to enhancers is most likely due to activation of this pathway upon loss of G9a. Nevertheless we performed DNAseI-qPCR hypersensitivity assay to address potential increased accessibility of the loci (Figure 14). However we did not observe changes at enhancers with increased eRNA expression in G9aKO EpiLCs. This result is in line with a lack of change to the H3K27ac at these loci (Figure 7). This further indicates that there is some level of uncoupling between enhancer activity and its accessibility during the dynamic process of inactivation.

Author response image 5.Bar plot showing DNAseI hypersensitivity of selected enhancers in d2 *G9a^F/-^ CreER^+ve^* EpiLCs treated with ethanol and tamoxifen.Shown is average from two independent biological repeats (+/- StDev).Procedure was performed as previously described (John et al., 2013)**DOI:**
http://dx.doi.org/10.7554/eLife.09571.036

4) The authors generate valuable H3K9me2 and H3K27me3 ChIP-seq datasets from epiblast cells of E6.25 embryos, but I feel they missed the opportunity to provide a deeper comparison of their results with the H3K9me2 and H3K27me3 patterns reported from the in vitro cellular models such as EpiLC or EpiSC. What are the major similarities and differences in patterns? A systematic (and to whatever extent possible, quantitative) comparison of H3K9me2 patterns across the genome in different cellular states would not only be broadly interesting for the community, but also relevant for the authors' own conclusions.

We have now included an in-depth comparison between the epiblast in vivo and EpiSC in vitro (Figure 4 and Figure 4—figure supplement 1). This includes the results from lcChIPseq for H3K9me2 and H3K27me3 modifications as well as WGBSeq dataset. Both cell types have strikingly similar distribution of H3K9me2 showing that accumulation of this modification occurs rapidly and efficiently both in vivo an in vitro. On the other hand, there is substantial redistribution of H3K27me3 during EpiSC derivation including loss of this modification from multiple promoters of germline regulators. We also show that self-renewing EpiSCs accumulate substantial levels of DNA methylation especially at regions where the polycomb mark is lost. This explains, at least in part, reduced developmental potential of EpiSCs for the germline fate due to stable epigenetic silencing of PGC genes.